# Disruption of TIGAR-TAK1 alleviates immunopathology in a murine model of sepsis

Dongdong Wang[1,2,3,8], Yanxia Li[1,4,8], Hao Yang [5,8], Xiaoqi Shen[1], Xiaolin Shi[1], Chenyu Li[1], Yongjing Zhang[1], Xiaoyu Liu[6], Bin Jiang[1], Xudong Zhu[1], Hanwen Zhang [1], Xiaoyu Li[1], Hui Bai[1], Qing Yang[1], Wei Gao [4,6], Fang Bai [5], Yong Ji [1,3], Qi Chen [1,2] ✉ & Jingjing Ben [1,7] ✉

Macrophage-orchestrated inflammation contributes to multiple diseases including sepsis. However, the underlying mechanisms remain to be defined clearly. Here, we show that macrophage TP53-induced glycolysis and apoptosis regulator (TIGAR) is up-regulated in murine sepsis models. When myeloid *Tigar* is ablated, sepsis induced by either lipopolysaccharide treatment or cecal ligation puncture in male mice is attenuated via inflammation inhibition. Mechanistic characterizations indicate that TIGAR directly binds to transforming growth factor β-activated kinase (TAK1) and promotes tumor necrosis factor receptor-associated factor 6-mediated ubiquitination and autophosphorylation of TAK1, in which residues 152-161 of TIGAR constitute crucial motif independent of its phosphatase activity. Interference with the binding of TIGAR to TAK1 by 5Z-7-oxozeaenol exhibits therapeutic effects in male murine model of sepsis. These findings demonstrate a non-canonical function of macrophage TIGAR in promoting inflammation, and confer a potential therapeutic target for sepsis by disruption of TIGAR-TAK1 interaction.

Inflammation is a defense system essential for survival. The inflammatory response can be triggered by invading pathogens or endogenous stress signals, leading to the clearance of infective agents and other inflammatory triggers. However, uncontrolled, excessive, and persistent inflammation is associated with a wide variety of diseases that dominate current morbidity and mortality worldwide[1,2]. The complex pathophysiological processes involved in the inflammatory response need for a deeper understanding of inflammation biology, which is necessary for the development of successful therapies targeting inflammation.

Inflammation involves multiple cell types, including myeloid cells, lymphocytes, and non-immune cells. The subcellular inflammatory processes are regulated by important subcellular compartment-specific macromolecules that afford the possibility of cell type-specific treatment by targeting their activity[3]. Macrophages are major innate immune cells resided in multiple organs and are functionally plastic to orchestrate inflammation in various diseases[4]. Upon recognizing pathogen-associated molecular patterns (PAMPs) or damage-associated molecular patterns (DAMPs), pattern recognition

[1]Department of Pathophysiology, Key Laboratory of Targeted Intervention of Cardiovascular Disease, Collaborative Innovation Center for Cardiovascular Disease Translational Medicine, Nanjing Medical University, Nanjing, China. [2]The Affiliated Suzhou Hospital of Nanjing Medical University, Suzhou Municipal Hospital, Gusu School, Nanjing Medical University, Nanjing, China. [3]Key Laboratory of Cardiovascular and Cerebrovascular Medicine, Nanjing Medical University, Nanjing, China. [4]The Affiliated Changzhou No. 2 People's Hospital of Nanjing Medical University, Changzhou Medical Center, Nanjing Medical University, Nanjing, China. [5]School of Life Science and Technology, and Shanghai Institute for Advanced Immunochemical Studies, ShanghaiTech University, Shanghai, China. [6]Jiangsu Key Lab of Cancer Biomarkers, Prevention and Treatment, School of Basic Medical Sciences, Nanjing Medical University, Nanjing, China. [7]The Affiliated Wuxi People's Hospital of Nanjing Medical University, Wuxi People's Hospital, Wuxi Medical Center, Nanjing Medical University, Nanjing, China. [8]These authors contributed equally: Dongdong Wang, Yanxia Li, Hao Yang. ✉e-mail: qichen@njmu.edu.cn; bjj@njmu.edu.cn

receptors (PRRs) are activated to produce various pro-inflammatory mediators, which are mediated via inflammatory signals including nuclear factor-κB (NF-κB), mitogen-activated protein kinase (MAPK), and Janus kinase-signal transducer and activator of transcription (JAK-STAT) signaling[3,5,6]. Changes in these signaling determine the magnitude and duration of the downstream response and, therefore, would be used as therapeutical potential for modulating inflammation. For example, transforming growth factor β-activated kinase (TAK1) acts as a pivotal kinase for promoting inflammation in macrophages. Its ubiquitination by E3 ubiquitin ligase tumor necrosis factor receptor-associated factor 6 (TRAF6) causes subsequent autophosphorylation and activates the inhibitor of κB (IκB) kinase (IKK), ultimately activating transcription factor NF-κB[5,7,8]. Suppression of TAK1 in macrophages is supposed to effectively ameliorate lethal infections such as sepsis[9].

Recently, metabolic flux has emerged as one of the important regulators for intracellular signaling cascades transduction[10,11]. Metabolic kinases do not only regulate energy production and catabolic and anabolic processes but also act as signaling molecules to regulate a variety of protein substrates and critical cellular processes including inflammation[12,13]. TP53-induced glycolysis and apoptosis regulator (TIGAR) is a downstream target gene of p53 and shares a similar structure with fructose-2,6-bisphosphatase (FBPase-2,6). It functions in hydrolyzing fructose-2,6-bisphosphate (Fru-2,6-P2) to fructose-6-phosphate (Fru-6-P), which inhibits glycolysis and increases the cellular NADPH level to activate the pentose phosphate pathway[14]. TIGAR has also been reported as a nonenzymatic function interacting with various proteins to regulate multiple cellular processes including mitochondrial homeostasis, cell viability, and chemotherapy resistance[15–18]. However, the role of TIGAR in human inflammatory diseases is not clear. Advances in understanding the protein phosphatase activity as TIGAR would pave a useful path for the development of specific interventions against inflammation.

In this study, we uncovered that TIGAR promotes inflammatory responses by activating the IKK-NF-κB signaling pathway in macrophages. The mechanism underlying it is independent of its phosphatase activity. Instead of, TIGAR directly binds to TAK1 and promotes TRAF6-mediated ubiquitination and autophosphorylation of TAK1. Interference with TIGAR-TAK1 interaction by 5Z-7-OX inhibited inflammation and alleviated murine sepsis induced by lipopolysaccharide (LPS) or cecal ligation puncture (CLP). Our findings demonstrate a noncanonical function of macrophage TIGAR in promoting inflammation. Disruption of TIGAR-TAK1 binding may be a promising therapeutic target against inflammatory diseases.

## Results
### Macrophage TIGAR exacerbates murine sepsis
The role of macrophage TIGAR in acute severe infection was examined by evaluating its response to the LPS-induced septic shock in male mice. We found that the expression of TIGAR was significantly up-regulated in F4/80+ cells isolated from both murine blood and spleen (Fig. 1a, b). When myeloid-specific Tigar knockout (MacKO, Tigar^flox/flox Lyz2-Cre^KI/KI) mice were generated by crossing Tigar^flox/flox mice with Lyz2-Cre^KI/KI mice (Fig. 1c–e and Fig. S1a), vitality and survival rate of LPS-administrated MacKO mice were significantly improved compared to those of littermates (Tigar^flox/flox Lyz2-Cre^WT/WT, MacWT mice) (Fig. 1f, g). Consistently, levels of important pro-inflammatory mediators in lung, liver, and plasma were also decreased in MacKO mice (Fig. 1h, i), reflecting an inhibitive effect of myeloid deficiency in TIGAR on LPS-induced acute inflammation. Of note, ablation of myeloid Tigar inhibited the expression of pro-inflammatory genes in spleen F4/80+ cells (Fig. 1j).

We also verified the role of macrophage TIGAR in murine CLP-induced sepsis. As expected, consistent phenotypic changes to those of LPS-inoculated MacKO mice were observed in the male CLP-treated MacKO mice (Fig. 1k–n). These results demonstrate a detrimental role

of macrophage TIGAR in murine sepsis, which would be irrelevant to the clotting system since no changes in tail bleeding time and activated partial thromboplastin time (APTT) were observed by ablation of myeloid Tigar (Fig. S1b, c).

### TIGAR promotes macrophage inflammatory response
Consistent with the above in vivo observations, treatment with LPS (Fig. 2a, b) induced upregulation of TIGAR either in cultured bone marrow-derived macrophages (BMDMs) or in RAW264.7 cells. mRNA sequencing (RNA-seq) revealed that ablation of TIGAR caused the downregulation of 131 genes and the upregulation of 22 genes in the LPS-treated BMDMs (Fig. S2a). These 131 downregulated genes are closely associated with the cellular responses to cytokines and immune-related processes (Fig. S2b,c). The pro-inflammatory character of macrophage TIGAR was further verified by measurements of pro-inflammatory mediators. We found that Tigar deficiency led to less production of pro-inflammatory mediators in either LPS-treated (Fig. 2d, e, Fig. S2c) or tumor necrosis factor (TNF)-α treated Tigar KO BMDMs (Fig. S2d). Inversely, TIGAR overexpression stimulated the LPS-induced overproduction of inflammatory mediators in macrophages (Fig. 2f, g). Together, these data reveal that TIGAR facilitates pro-inflammatory responses in macrophages.

### TIGAR stimulates inflammation by activating IKK-NF-κB signaling in macrophages
To gain molecular insights into the macrophage inflammatory response that is mitigated by Tigar deletion, we conducted PANTHER enrichment analysis. The use of RNA-Seq data of LPS-treated macrophages revealed that Tigar deficiency affected the interleukin signaling, chemokine and cytokine signaling, and toll-like receptor (TLR) signaling pathways (Fig. S3a). Interestingly, all these pathways converge in NF-κB signaling-mediated inflammatory response in sepsis[19,20]. Indeed, Tigar ablation inhibited phosphorylation of IKK and p65, IκB-α degradation, and nuclear translocation of p65 in LPS-treated BMDMs (Fig. 3a–c) as well as phosphorylation of p65 in TNF-α-treated BMDMs (Fig. S3b). These in vitro findings were confirmed by in vivo observations that knockout of myeloid Tigar inhibited phosphorylation of IKK and p65, and IκB-α degradation in septic murine lung tissues (Fig. S3c).

In contrast to the observations from the Tigar knockout experiments, we found that TIGAR overexpression in cultured RAW264.7 cells significantly activated IKK-NF-κB signaling (Fig. 3d, e) and facilitated inflammatory response (Fig. 3f). The activating effect of TIGAR on IKK-NF-κB pathway was further determined by use of a specific NF-κB pathway inhibitor, Bay11-7082. Figure 3f showed that inhibition of the NF-κB pathway dramatically inhibited the production of pro-inflammatory mediators in cells even in the setting of TIGAR overexpression. Thus, TIGAR may aggravate the LPS-triggered inflammatory response via activating IKK-NF-κB signaling in macrophages.

### TIGAR activates the NF-κB signaling pathway by promoting signal transducer TAK1 phosphorylation in macrophages
TIGAR is a phosphatase functioning in the generation of antioxidant NADPH by activating the pentose phosphate pathway[14]. However, we found that Tigar ablation decreased reactive oxygen species (ROS) generation in the LPS-treated BMDMs (Fig. S4a, b), implying a phosphatase-independent function of TIGAR in the LPS-agitated cells. This speculation was verified by the mutation of the three catalytic pocket residues (His11, Glu102, and His198) of TIGAR to alanine[21]. Interestingly, transfection of the TIGAR mutant (TMU) into RAW264.7 cells did not affect the stimulative effect of TIGAR on macrophage inflammatory response (Fig. S4c, d), demonstrating a phosphatase-free mechanism underlying the TIGAR-mediated NF-κB signaling activation.

TLR4 pathway is known as the ubiquitous key modulator for inflammatory pathways including NF-κB signaling in cells[5]. We

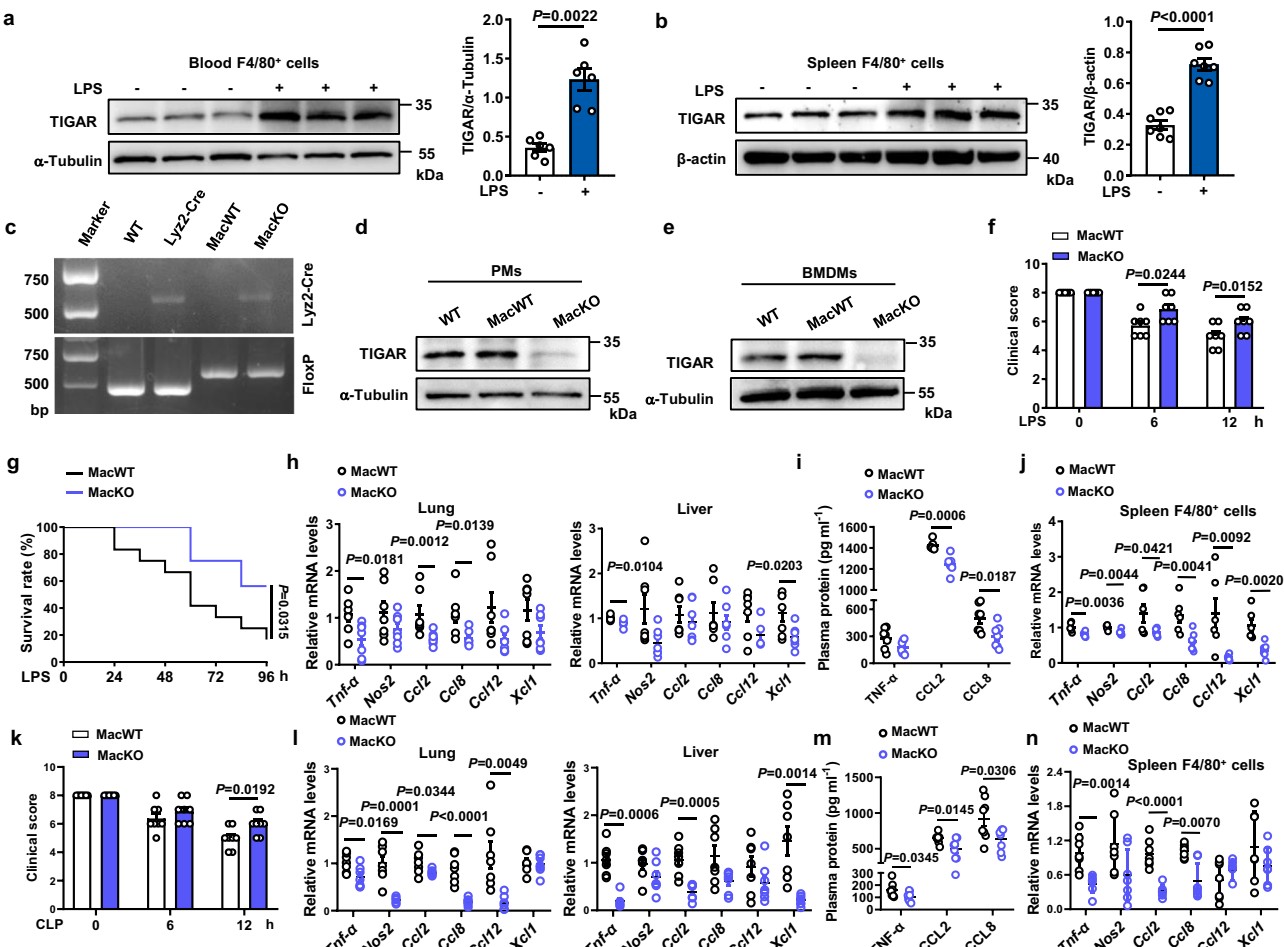

**Fig. 1 | Ablation of *Tigar* ameliorates inflammation in murine sepsis. a, b** Male C57BL/6 J mice were intraperitoneally injected with LPS (10 mg kg⁻¹) or phosphate-buffered saline (PBS). The mice were euthanized 12 h later. Western blot analysis of TIGAR expression in F4/80⁺ cells isolated from both blood (*n* = 6) (**a**) and spleen (*n* = 7) (**b**). **c** Representative DNA gel image of WT, *Lyz2*-Cre, MacWT (*Tigar*flox/flox), and MacKO (*Tigar*flox/flox*Lyz2*-CreKI/KI) mice by PCR amplification. The experiment was repeated three times independently with similar results. **d, e** Western blot of TIGAR in murine peritoneal macrophages (PMs) (**d**) and BMDMs (**e**). Blot assay was repeated three times independently with similar results. **f** Mice were intraperitoneally injected with LPS (10 mg kg⁻¹) and euthanized 12 h later. Clinical score was determined at the indicated time (*n* = 7). **g** Murine survival rate was determined during 96 h of LPS challenge (*n* = 12). **h** mRNA levels of pro-inflammatory genes in the murine lung (left) and liver (right) after 12 h of LPS challenge (*n* = 7). **i** Plasma levels of TNF-α, C–C motif ligand-2 (CCL2) and C–C motif ligand-8 (CCL8) in septic mice (*n* = 7). **j** mRNA levels of pro-inflammatory genes in the spleen F4/80⁺ cells from MacWT (*n* = 6) and MacKO (*n* = 7) mice. **k** Male mice were induced CLP sepsis and euthanized 12 h later. Clinical score was determined at the indicated time (*n* = 8). **l** mRNA levels of pro-inflammatory genes in the murine lung (left) and liver (right) after 12 h of CLP surgery (*n* = 7). **m** Plasma levels of TNF-α, CCL2 and CCL8 in CLP septic mice (*n* = 8). **n** mRNA levels of pro-inflammatory genes in the spleen F4/80⁺ cells from MacWT and MacKO (*n* = 7) mice. Data are expressed as mean ± SEM. **a** Two-tailed Mann–Whitney U test. **b, f, k** Two-tailed Student *t*-test. **g** Log-rank (Mantel–Cox) test. **h–j, l–n** Two-tailed Student *t*-test, Two-tailed Mann–Whitney U test. Source data are provided as a Source Data file.

searched for the potential interactors in the TLR4 pathway with TIGAR. When TLR4, myeloid differentiation primary response gene 88 (MyD88), interleukin 1 receptor-associated kinase 1 (IRAK1), TRAF6, TGF-β activated kinase 1 binding protein 1/2 (TAB1/2), and TAK1 were co-transfected with exogenous TIGAR into HEK293 cells respectively, only TRAF6 and TAK1 could directly bind with TIGAR (Fig. S5a–f, Fig. 4a, b). However, the interaction between TRAF6 and TIGAR in the transfected cells did not influence the TRAF6 oligomerization and ubiquitination, the key events in activating NF-κB signaling[22] (Fig. S5g, h). Therefore, we did not further investigate the interaction between TIGAR and TRAF6 in the subsequent experiments. Differing from TRAF6, TAK1 did not only exhibit a strong interaction with TIGAR, its endogenous co-precipitation with TIGAR in BMDMs (Fig. 4c) was more evident upon LPS stimulation (Fig. 4d). The co-localization of endogenous TIGAR with TAK1 was also evidenced in macrophages by immunofluorescence staining (Fig. 4e). Meanwhile, TIGAR could not bind with any subunit of the linear ubiquitination assembly

complex (LUBAC) in macrophages (Fig. S6a), though interaction between TIGAR and the HOIP subunit of LUBAC was found in adipocytes[23].

TAK1 is a serine and threonine kinase activated by ubiquitination and subsequent autophosphorylation[8,24,25]. We found that TAK1 kinase catalytic activity was not directly affected by the presence of TIGAR in vitro (Fig. S6b). However, the LPS-induced phosphorylation of TAK1 was inhibited by *Tigar* ablation in BMDMs (Fig. 4f). This reaction pattern was also found in the LPS-administrated MacKO lung tissues (Fig. S6c). Treatment with (5Z)-7-Oxozeaenol (5Z-7-OX), a TAK1 inhibitor[26], blocked the TIGAR-boosted activation of IKK-NF-κB signaling (Fig. 4g, h) and production of pro-inflammatory mediators (Fig. 4i) in macrophages. Consistently, alternative TAK1 inhibitor Takinib also showed an inhibitory effect on the TIGAR-induced macrophage inflammation (Fig. S6d). As such, TAK1 may act as an important signal transducer contributing to the TIGAR-activated NF-κB signaling by its phosphorylation.

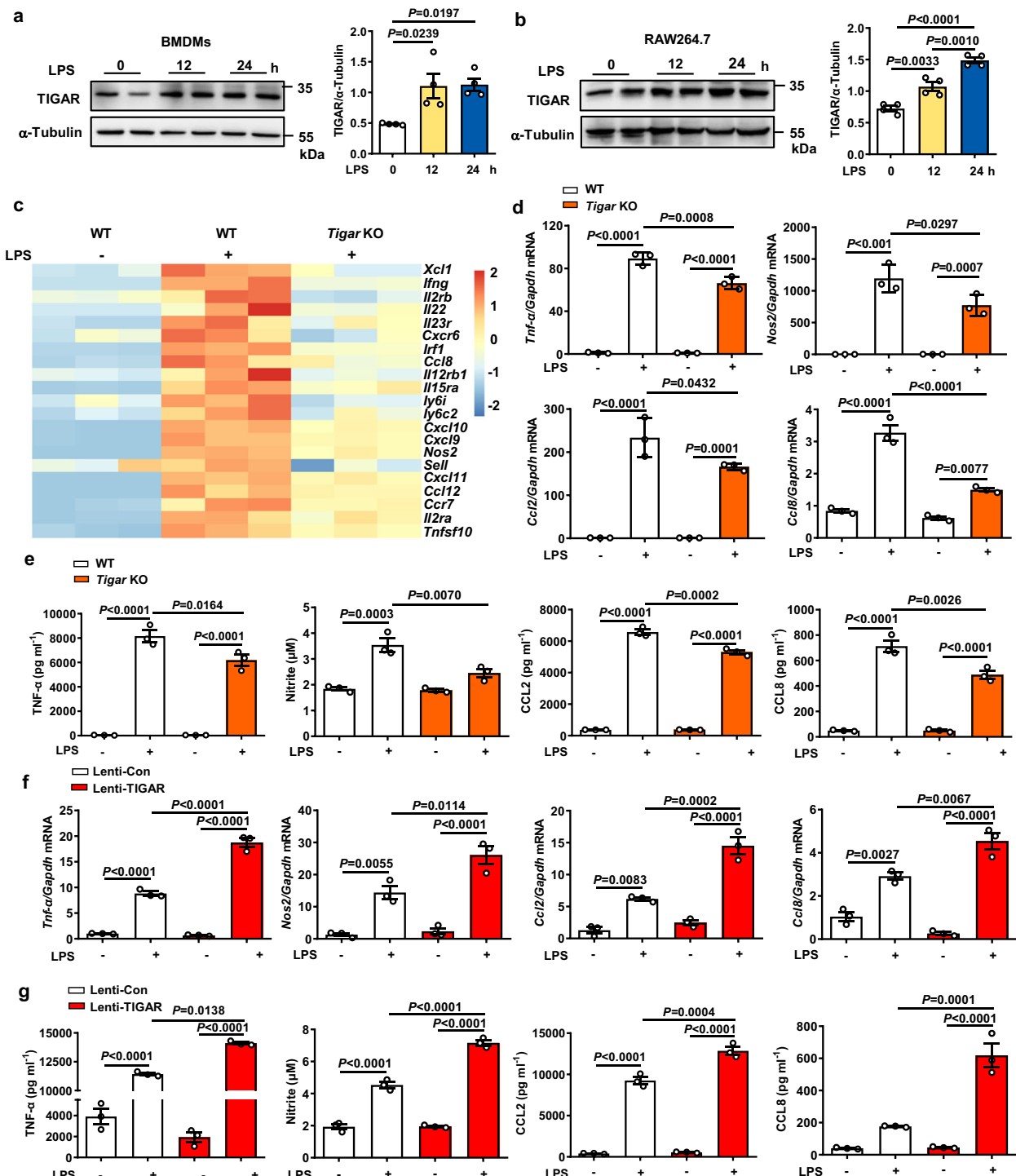

**Fig. 2 | TIGAR activates pro-inflammatory responses in macrophages.**
**a**, **b** Western blot analysis of TIGAR expression in BMDMs (**a**) and RAW264.7 cells (**b**) stimulated with LPS (100 ng ml⁻¹) for indicated times, $n = 4$ samples. **c** Heatmap of the selected pro-inflammatory genes transcript expressions in *Tigar* KO BMDMs stimulated with LPS for 8 h compared to WT controls. **d**, **e** BMDMs isolated from WT and *Tigar* KO mice were stimulated with LPS for 8 h. mRNA levels of pro-inflammatory genes including *Tnf-α*, *Nos2*, *Ccl2*, and *Ccl8* were assessed ($n = 3$) (**d**). TNF-α, CCL2, and CCL8 levels in culture media were determined using ELISA.

Nitrite, the end product of NO metabolism, was measured by Griess reagent ($n = 3$) (**e**). **f**, **g** RAW264.7 cells were transfected by Lenti-viruses expressing control Flag (Lenti-Con) and Flag-TIGAR (Lenti-TIGAR) for 72 h followed by LPS treatment for 12 h. mRNA levels of the pro-inflammatory genes in RAW264.7 cells ($n = 3$) (**f**). TNF-α, CCL2, and CCL8 levels in culture media were determined using ELISA. Nitrite, the end products of NO metabolism, measured by Griess reagent ($n = 3$) (**g**). **a**, **b**, **d**–**g** One-way ANOVA followed by the Bonferroni test. Source data are provided as a Source Data file.

## TIGAR promotes TAK1 ubiquitination by inducing interaction between TRAF6 and TAK1

Ubiquitination is essential for the subsequent autophosphorylation and activation of TAK1[8,24,25], which was positively regulated by TIGAR in

BMDMs (Fig. 5a) as well as in transfected HEK293 cells (Fig. 5b). Importantly, TIGAR overexpression preferentially increased the TAK1-Ub conjugates at K63 instead of K48 (Figs. 5c, d, and S7a). TAK1 ubiquitination is regulated by TAB1, TRAF6 E3 ligase, and deubiquitinating

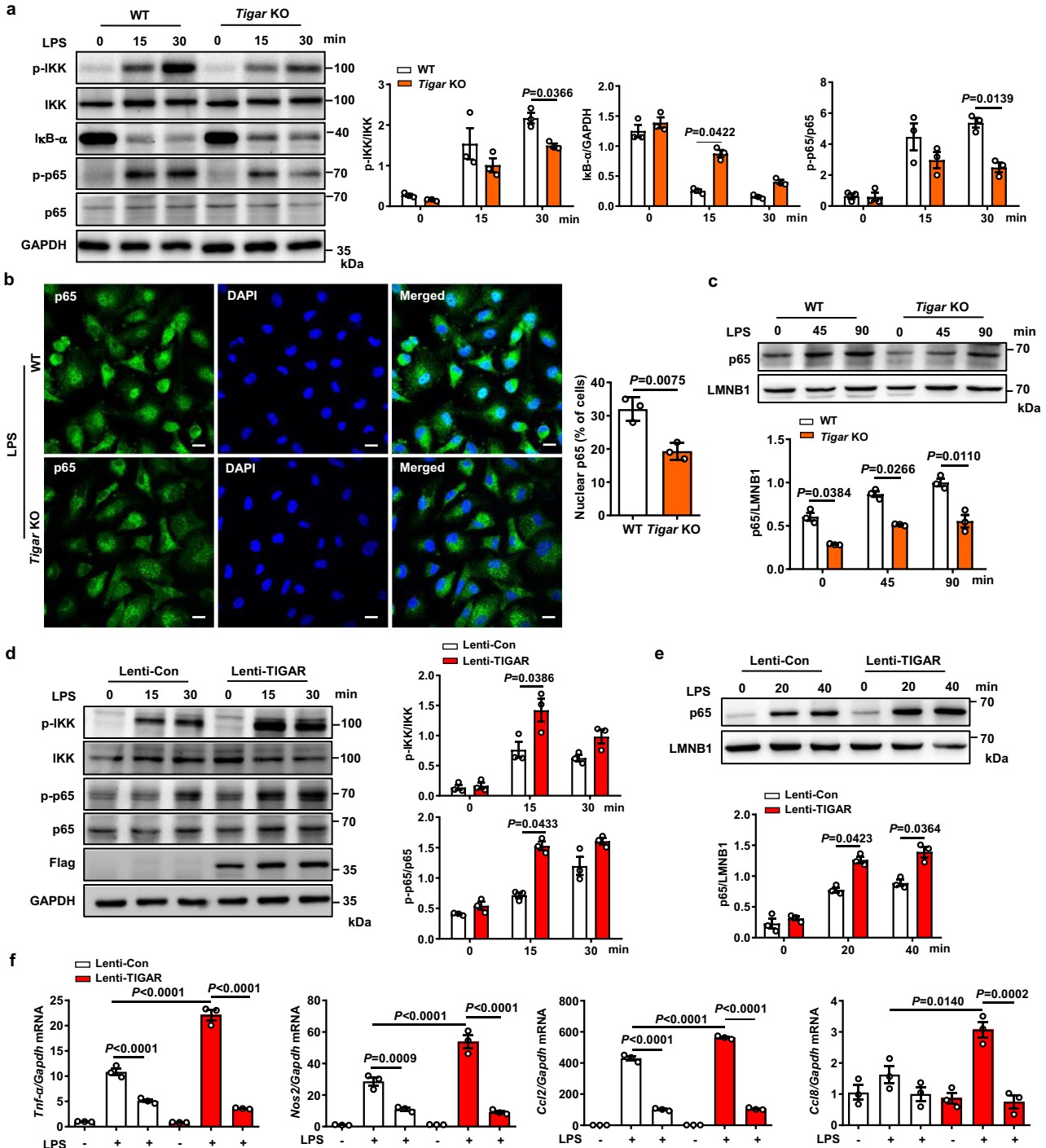

**Fig. 3 | TIGAR-deficient reduces pro-inflammatory gene expression by inhibiting IKK-NF-κB signaling in macrophages. a** Western blot analysis of cell lysates from WT and *Tigar* KO BMDMs treated with LPS for the indicated times using the indicated antibodies (*n* = 3). **b** PMs isolated from WT and *Tigar* KO mice were treated with LPS for 3 h. Immunofluorescent staining of p65 (green) and DAPI (blue) (*n* = 3). Scale bars, 10 µm. **c** BMDMs isolated from WT and *Tigar* KO mice were treated with LPS for the indicated times. Western blot analysis of p65 subcellular distribution in the nucleus (*n* = 3). **d** RAW264.7 cells were transfected by Lenti-Con or Lenti-TIGAR for 72 h followed by LPS treatment for indicated time. Western blot analysis of cell lysates using the indicated antibodies (*n* = 3). **e** RAW264.7 cells were

transfected by Lenti-Con or Lenti-TIGAR for 72 h followed by LPS treatment for indicated time. Western blot of p65 subcellular distribution in the nucleus from RAW264.7 cells treated with LPS for indicated times (*n* = 3). **f** RAW264.7 cells were transfected by Lenti-Con or Lenti-TIGAR for 72 h followed by LPS treatment with or without NF-κB pathway inhibitor BAY11-7082 for 12 h. mRNA levels of *Tnf-α*, *Nos2*, *Ccl2*, and *Ccl8* were detected in the indicated groups (*n* = 3). Data are expressed as mean ± SEM. **a–e** Two-way ANOVA followed by the Bonferroni test. **b** Two-tailed Student *t*-test. **f** One-way ANOVA followed by the Bonferroni test. Source data are provided as a Source Data file.

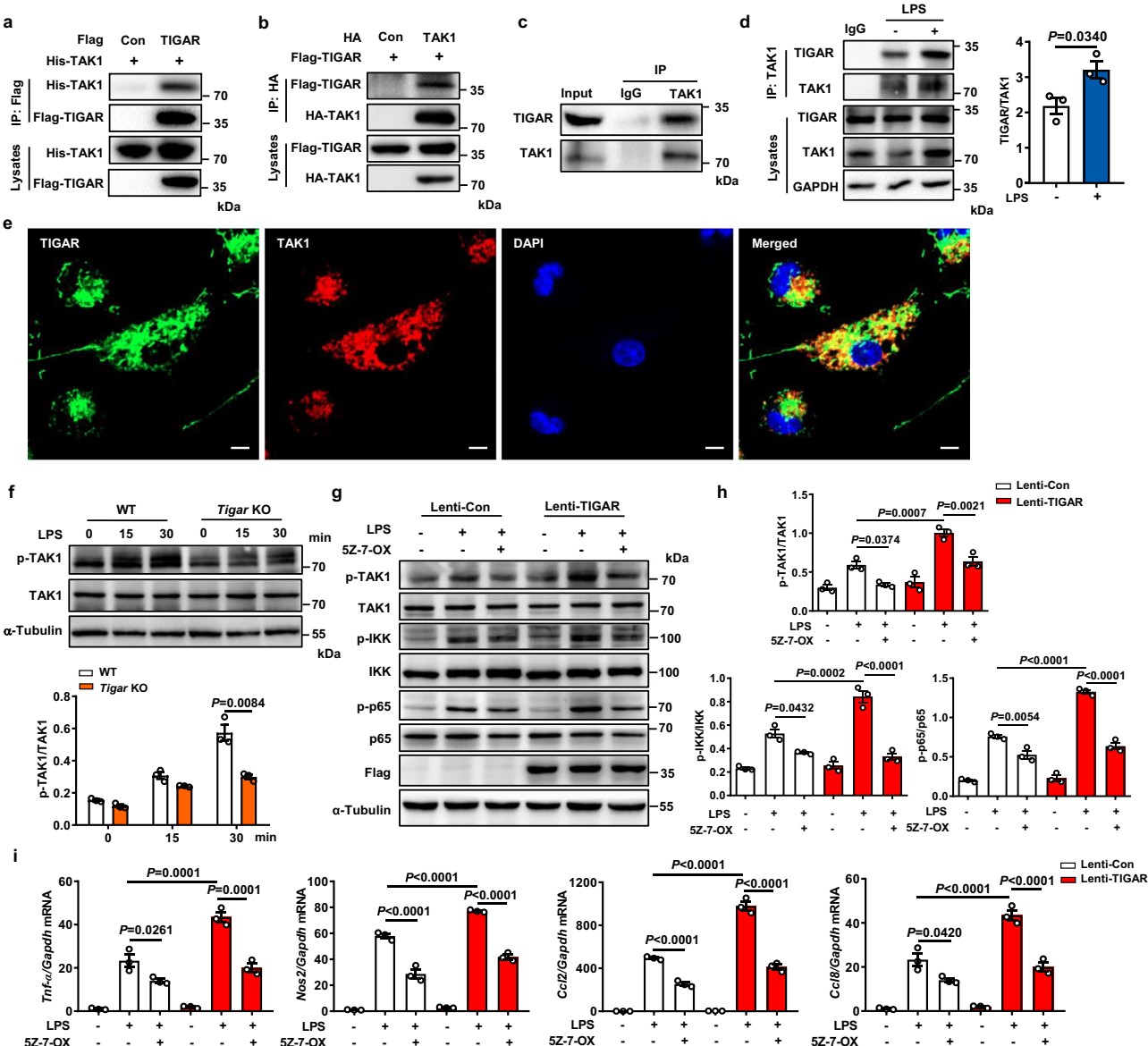

**Fig. 4 | TIGAR ablation inhibits TAK1 activation. a** HEK293 cells were transfected by His-TAK1, Flag-TIGAR, or Flag empty vector. Co-IP and western blot of TAK1 and TIGAR in the transfected cells. **b** HEK293 cells were transfected by Flag-TIGAR, HA-TAK1, or HA empty vector. Co-IP and western blot of TIGAR and TAK1 in the transfected cells. **c** Co-IP and western blot of endogenous TIGAR and TAK1 in BMDMs. **d** Co-IP and quantifications of endogenous TIGAR and TAK1 in BMDMs stimulated by LPS. $n = 3$ independent experiments. **e** Confocal microscopic images of TIGAR (green) and TAK1 (red) in PMs. Scale bars, 5 μm. Images were representative images from three independent experiments. **f** Western blot of TAK1 in BMDM lysates from WT and *Tigar* KO mice treated by LPS for the indicated times using the indicated antibodies, $n = 3$ samples. **g** RAW264.7 cells were transfected

with Lenti-Con or Lenti-TIGAR for 72 h followed by LPS treatment with or without TAK1 inhibitor 5Z-7-OX (100 nM) for 30 min. Western blot analysis of cell lysates using the indicated antibodies. **h** Quantifications of the phosphorylation and total TAK1, IKK, and p65 in the indicated groups, $n = 3$ samples. **i** mRNA levels of pro-inflammatory genes in RAW264.7 cells followed by LPS treatment with or without TAK1 inhibitor 5Z-7-OX (100 nM) for 12 h ($n = 3$). Data are expressed as mean ± SEM. **d** Two-tailed Student *t*-test. **f** Two-way ANOVA followed by the Bonferroni test. **h**, **i** One-way ANOVA followed by the Bonferroni test. All blot assays were repeated three times independently with similar results. Source data are provided as a Source Data file.

enzymes USP4/18[8,27,28]. We found that TIGAR overexpression promoted the interaction of TAK1 with TRAF6 (Fig. 5e) but not with TAB1, USP4 or USP18 (Fig. S7b–d). Conversely, TIGAR deficiency inhibited the LPS-induced interaction between TAK1 and TRAF6 (Fig. 5f, g). As an E3 ligase, TRAF6 may catalyze TAK1 ubiquitination by binding with it. This was verified by the observation that the dominant-negative mutant of TRAF6 (TRAF6-DN) did not only block the ubiquitination of TAK1 but also inhibited the TIGAR-potentiated ubiquitination of TAK1 (Fig. 5h), suggesting a bridge role of TRAF6 in TIGAR-triggered TAK1 ubiquitination.

Assembly of the TAK1-TRAF6 complex is facilitated by TAB2/3[29,30]. We found that the TAB2/3 knockdown blocked the TAK1-TRAF6 complex formation, which was reversed by TIGAR overexpression (Fig. S8a). Co-IP assays showed that TIGAR, TAK1, and TRAF6 interacted with each other and might form a tripolymer (Fig. 5i, j). When a set of truncated TRAF6 and TAK1 constructs were tested for reactivity (Fig. S8b), TAK1 (1–300) directly interacted with the TRAF6 (332–530) fragment (Fig. S8c, d), which was consistent with the previous report[31]. Interestingly, these two fragments could also be recognized by TIGAR (Fig. 5k, l). Furthermore, TMU interacted with either TAK1 or TRAF6 as

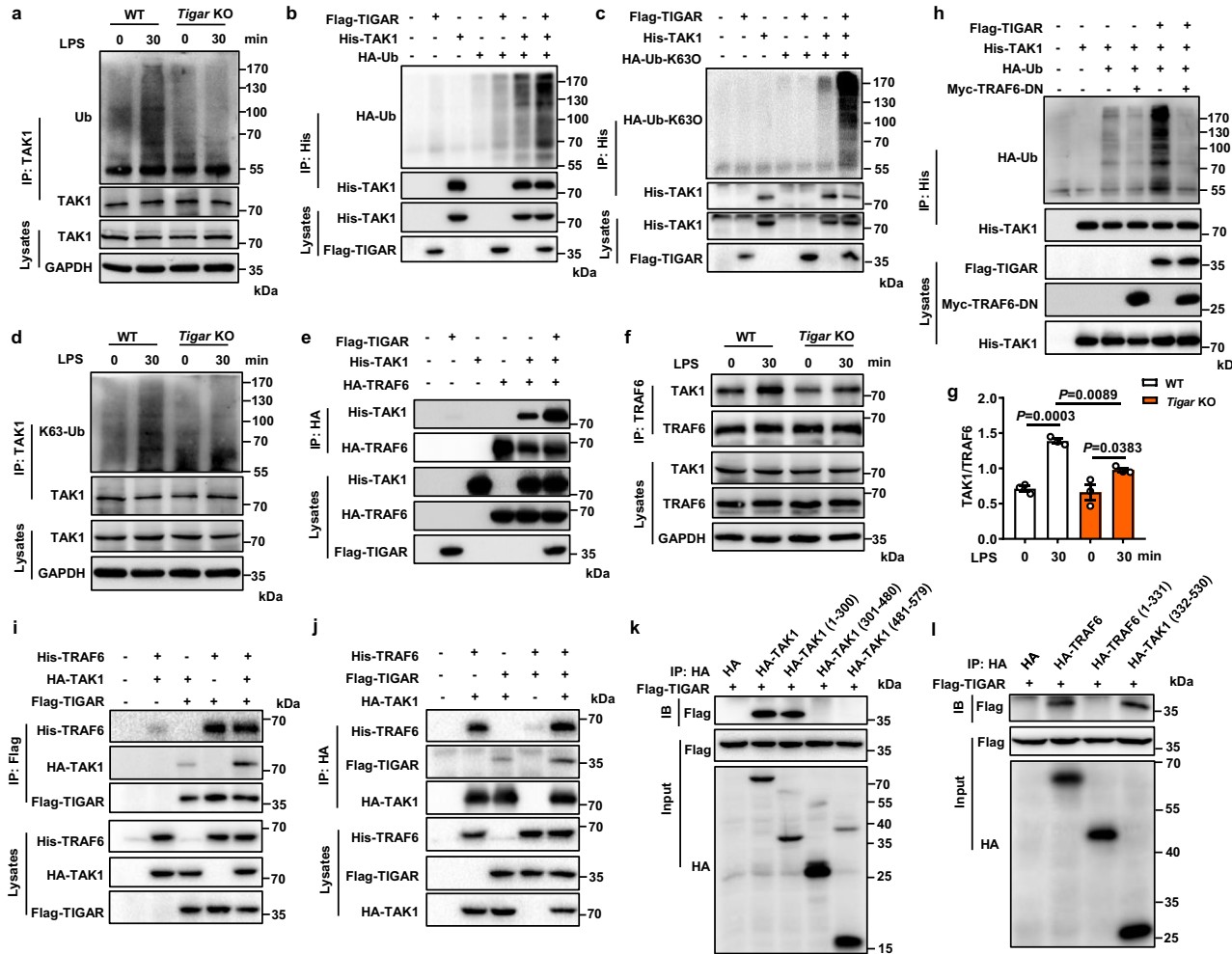

**Fig. 5 | TIGAR promotes TAK1 K63 ubiquitination mediated by E3 Ligase TRAF6. a** BMDMs isolated from WT and *Tigar* KO mice were stimulated with LPS for 30 min and subjected to Co-IP with anti-TAK1 antibody followed by western blot with anti-Ub antibody. **b** HEK293 cells were transfected by His-TAK1, HA-Ub and Flag-TIGAR plasmids. Co-IP and western blot of the ubiquitination of TAK1. **c** HEK293 cells were transfected by His-TAK1, HA-Ub-K63O and Flag-TIGAR plasmids. Co-IP and western blot of K63 ubiquitination of TAK1. HA-Ub-K63O indicates ubiquitin in which all lysines except K63 were mutated. **d** BMDMs isolated from WT and *Tigar* KO mice were stimulated with LPS for 30 min and subjected to Co-IP with anti-TAK1 antibody followed by western blot with anti-K63-Ub antibody. **e** HEK293 cells were transfected by His-TAK1, HA-TRAF6 and Flag-TIGAR plasmids. Co-IP and western blot of the complex formation between TAK1 and TRAF6. **f** BMDMs isolated from WT and *Tigar* KO mice were stimulated with LPS for 30 min and subjected to Co-IP with anti-TRAF6 antibody followed by western blot with anti-TAK1 antibody. **g** Quantifications of the effect of TIGAR on the complex formation between TAK1 and TRAF6 (*n* = 3). **h** HEK293 cells were transfected by indicated plasmids and TAK1 ubiquitination was detected by western blot. **i, j** Co-IP and western blot of the interactions between TIGAR, TAK1, and TRAF6 in HEK293 cells transfected by indicated plasmids. **k** HEK293 cells were transfected by Flag-TIGAR and HA-tagged TAK1 fragments. Co-IP and western blot of the binding between TIGAR and TAK1 fragments. **l** HEK293 cells were transfected by Flag-TIGAR and HA-tagged TRAF6 fragments. Co-IP and western blot of the binding between TIGAR and TRAF6 fragments. Data are expressed as mean ± SEM. **g** One-way ANOVA followed by the Bonferroni test. All blot assays were repeated three times independently with similar results. Source data are provided as a Source Data file.

efficiently as WT TIGAR (Fig. S9a, b). TMU could also promote TAK1 and TRAF6 complex formation (Fig. S9c) and ubiquitination of TAK1 (Fig. S9d). Therefore, TIGAR may activate NF-κB signaling via a phosphatase activity-free manner.

## Residues 152–161 of TIGAR is crucial for TAK1 activation

To decipher the molecular basis for TIGAR mediating TAK1 activation, we firstly compared the reactivity of truncated fragments TIGAR (1–210) and TIGAR (211–269) for binding with TAK1[15]. Unexpectedly, TIGAR (1–210) instead of TIGAR (211–269) could bind with either TAK1 or TRAF6 (Fig. 6a, b). In agreement, TIGAR (1–210) but not TIGAR (211–269) could promote the interaction between TAK1 and TRAF6 and the ubiquitination of TAK1 (Fig. 6c, d). Consequently, TIGAR (1–210) exacerbated the LPS-induced production of inflammatory mediators (Fig. 6e).

Next, we performed a series of comprehensive computational simulations to analyze the TIGAR-TAK1 complex formation. An initial dimer complex was generated by combining several computational methods, including protein–protein docking, dimer classification, and molecular mechanics/generalized born surface area (MM/GBSA) refinement package. We then employed three times 400 ns-long molecular dynamics (MD) simulations to optimize the initial formation. The residues within the entire system and interface reached states of convergences after simulations of 200 ns (Fig. S10a). After performing an MD simulation, the cavities presented on the initial protein interface were eliminated, resulting in a more tightly fitting and complementary interface (Fig. S10b). The Gibbs free energy landscape (FEL) plot revealed the deepest energy well, from which we derived the predictive binding mode between TIGAR and TAK1 (Fig. 6f). Consistent with previous truncation experiments, this computational model

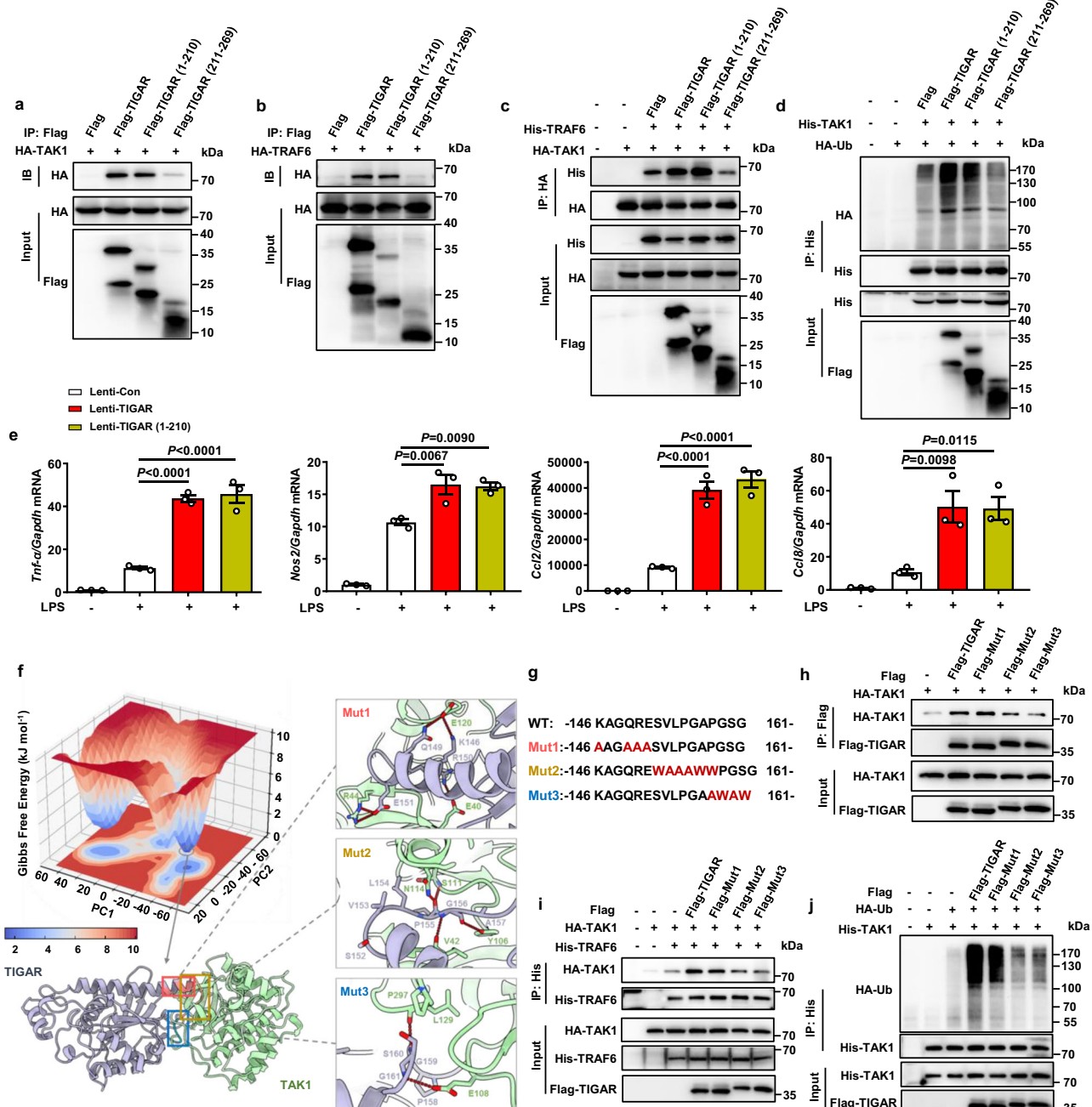

**Fig. 6 | Residues 152–161 of TIGAR is pivotal in activating TAK1 activity.**
**a**, **b** HEK293 cells were transfected by HA-TAK1, HA-TRAF6 and Flag-TIGAR fragments. Co-IP and western blot of the binding between TIGAR fragments and TAK1 (**a**) or TRAF6 (**b**). **c** Co-IP and western blot of the complex formation between TAK1 and TRAF6 after transfected by Flag-TIGAR fragments. **d** Co-IP and western blot of the ubiquitination of TAK1 after transfected by Flag-TIGAR fragments. **e** RAW264.7 cells were transfected by Lenti-Con, Lenti-TIGAR, and Lenti-TIGAR (1–210) for 72 h followed by LPS treatment for 12 h. mRNA levels of *Tnf-a*, *Nos2*, *Ccl2*, and *Ccl8* in RAW264.7 cells, *n* = 3 samples. **f** Gibbs free energy landscape of the first two principal components (PCs) generated from MD simulations for the binding between TAK1 and TIGAR, where the dark blue color areas indicate lower energetic conformations. The lowest potential well highlighted with a gray circle. According to the TAK1 (pale green)-TIGAR (purple) binding mode extracted from the lowest

potential well, the details of the critical molecular interaction between the two proteins have been zoomed in. **g** Wild type and three sets of designed mutant sequences of 146–161 residues of TIGAR for validating the predicted binding mode. **h** HEK293 cells were transfected by HA-TAK1 and Flag-tagged TIGAR mutants. Co-IP and western blot of the binding motif between TIGAR and TAK1. **i** HEK293 cells were transfected by HA-TAK1, His-TRAF6, and Flag-tagged TIGAR mutants. Co-IP and western blot of the complex formation between TAK1 and TRAF6 after transfected by Flag-tagged TIGAR mutants. **j** HEK293 cells were transfected by His-TAK1, HA-Ub and Flag-tagged TIGAR mutants. Co-IP and western blot of the ubiquitination of TAK1 after transfected by Flag-tagged TIGAR mutants. Data are expressed as mean ± SEM. **e** One-way ANOVA followed by the Bonferroni test. All blot assays were repeated three times independently with similar results. Source data are provided as a Source Data file.

showed that residues 211–269 of TIGAR were not involved in the interaction between TIGAR and TAK1 (Fig. S10c). In the residues 1–210 region, residues 146–161 were mainly involved in the interaction by the loop structure of residues 152–161 dynamically interacting with the

ATP-binding domain of TAK1, which primarily involved hydrophobic interactions and backbone hydrogen bonds. In addition, residues 146–151 of TIGAR appeared on the protein surface and formed some hydrogen bonds with TAK1 (Fig. 6f).

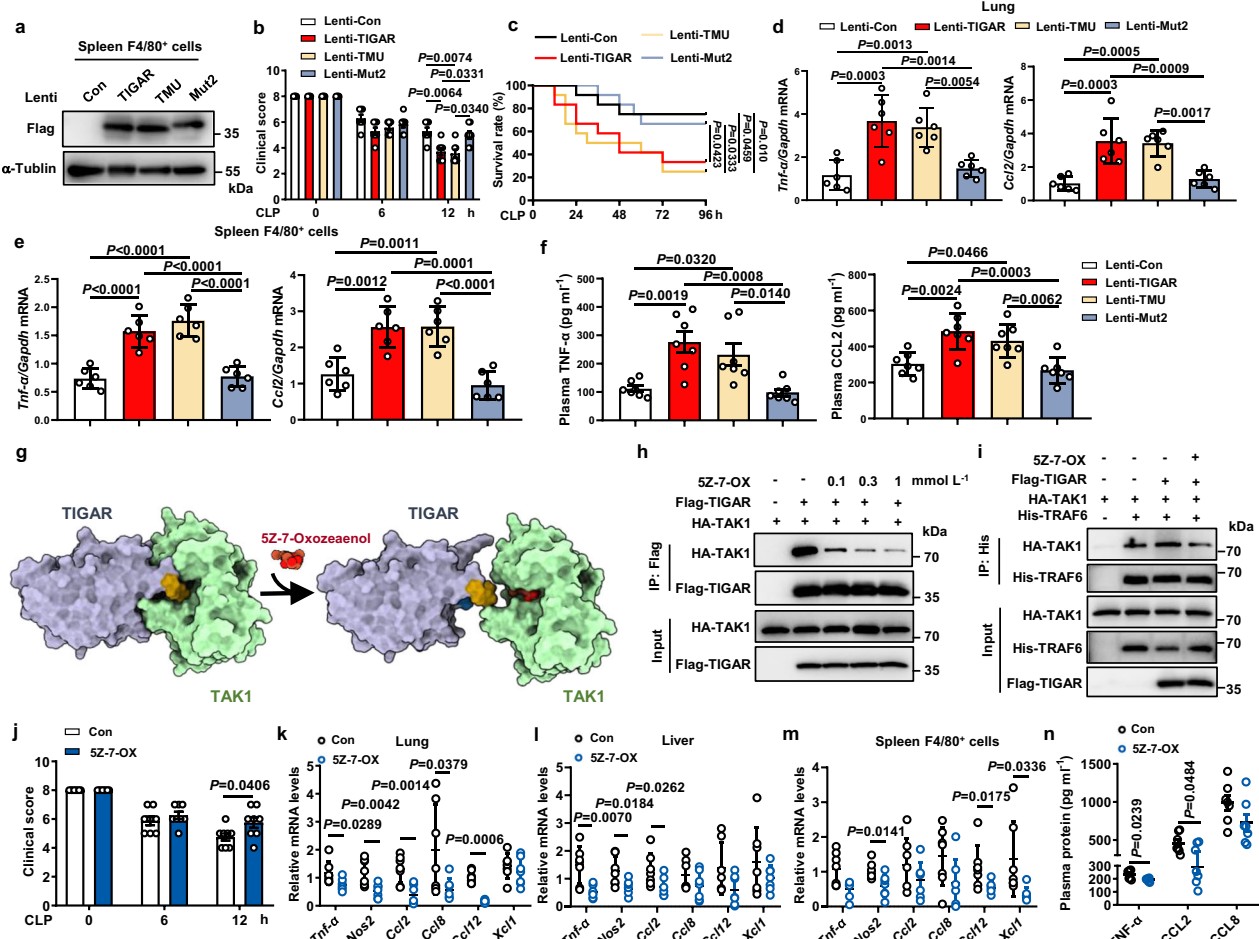

**Fig. 7 | Inhibiting TIGAR-TAK1 interaction abolishes murine sepsis.** *Tigar* KO mice were infected with myeloid-specific CD11b promoter-driven lentivirus (pCDH-CD11b-T2A-copGFP) encoding Flag-TIGAR, Flag-TMU, or Flag-Mut2, respectively. After 4 days the CLP sepsis model was generated. **a** Western blot of Flag-TIGAR, Flag-TMU and Flag-Mut2 expression in spleen F4/80⁺ cells. **b** Clinical score of mice was analyzed (*n* = 7). **c** Murine survival rate was determined during 96 h of CLP challenge (*n* = 12). **d, e** mRNA levels of pro-inflammatory genes in lung (**d**) (*n* = 6), and spleen F4/80⁺ cells (**e**) (*n* = 6). **f** Plasma concentrations of TNF-α, and CCL2 in mice (*n* = 7). **g** Schematic diagram of the mechanism underlying 5Z-7-OX competing with TIGAR for binding to TAK1. Yellow structure: residues 152–157 of TIGAR; Blue structure: residues 158–161 of TIGAR. **h** HEK293 cells were transfected by HA-TAK1 and Flag-TIGAR plasmids. Co-IP and western blot of TIGAR-TAK1 complex formation in HEK293 cells incubated with 0.1, 0.3, or 1 mM 5Z-7-OX for 12 h. **i** HEK293 cells

were transfected by HA-TAK1, His-TRAF6 and Flag-TIGAR plasmids and treated with or without 0.1 mM 5Z-7-OX. Co-IP and western blot of TAK1-TRAF6 complex formation in HEK293 cells. Male C57BL/6J mice were intraperitoneally injected with 5Z-7-OX or DMSO (Con). After 1 h, mice were induced with CLP sepsis and euthanized 12 h later. **j** Clinical score of mice was analyzed (*n* = 8). **k–m** mRNA levels of pro-inflammatory genes in lung (**k**) (*n* = 7), liver (**l**) (*n* = 7), and spleen F4/80⁺ cells (**m**) (*n* = 7). **n** Plasma concentrations of TNF-α, CCL2 and CCL8 in Con and 5Z-7-OX (*n* = 7) treated mice. Data are expressed as mean ± SEM. **b, d–f** One-way ANOVA followed by the Bonferroni test. **c** Log-rank (Mantel–Cox) test. **j** Two-tailed Student *t*-test. **k–n** Two-tailed Student *t*-test, Two-tailed *t*-test with Welch correction, or two-tailed Mann–Whitney U test. Source data are provided as a Source Data file.

To verify these speculations from computational model, we designed mutant TIGAR, denoted as Mut1, focusing on the residues 146–151 of TIGAR (Fig. 6g). The region of residues 152–161 was further divided into two sub-regions for the mutant designing, one for residues 152–157 (Mut2) and the other for residues 158–161 (Mut3) (Fig. 6g). We found that Mut3 had an attenuated binding ability with TAK1, TRAF6-TAK1 complex formation, and TAK1 ubiquitination compared with WT TIGAR (Fig. 6h–j). These results may be caused by disruption of the hydrogen bond between G161 of TIGAR and E108 of TAK1 as well as the introduction of steric hindrance by mutations of G159W and G161W. Mut2 had comparable effects as Mut3 on the interaction between TIGAR and TAK1 (Fig. 6h–j). This may be due to the disruption of hydrophobic interactions by mutations of V153A, L154A, and P155A, steric hindrance by mutations of S152W, G156W, and A157W, as well as the disruption of main chain hydrogen bonds. Mut1, mutations of residues 146–151, had no obvious impacts on the interaction between TIGAR and TAK1. This diminished effect could be

attributed to the fact that the disruption of exposed side chain hydrogen bonds in solution did not significantly influence the extensive and planar protein–protein interaction interface, thereby allowing for a relatively preserved binding affinity between the proteins. As such, these results highlight the pivotal role of TIGAR residues 152–161 in the formation of the TIGAR-TAK1 complex and subsequent TAK1 ubiquitination.

To further understand the mechanism of TIGAR activating TAK1 in vivo, we constructed myeloid-specific CD11b promoter-driven lentivirus encoding Flag-TIGAR, Flag-TMU, or Flag-Mut2, respectively. After 4 days of lentivirus infection in *Tigar* KO mice, the CLP septic model was generated. FACS analysis showed that an average of 37% of CD45⁺CD11b⁺ myeloid cells in blood and spleen expressed the transfected Flag-TIGAR, TMU, or Mut2 (Figs. S11a–c, S13). Western blot revealed that the transfected Flag-TIGARs were expressed in spleen F4/80⁺ macrophages (Fig. 7a) instead of lung epithelial cells and hepatocytes (Fig. S11d), respectively. Immunohistochemical (IHC) staining

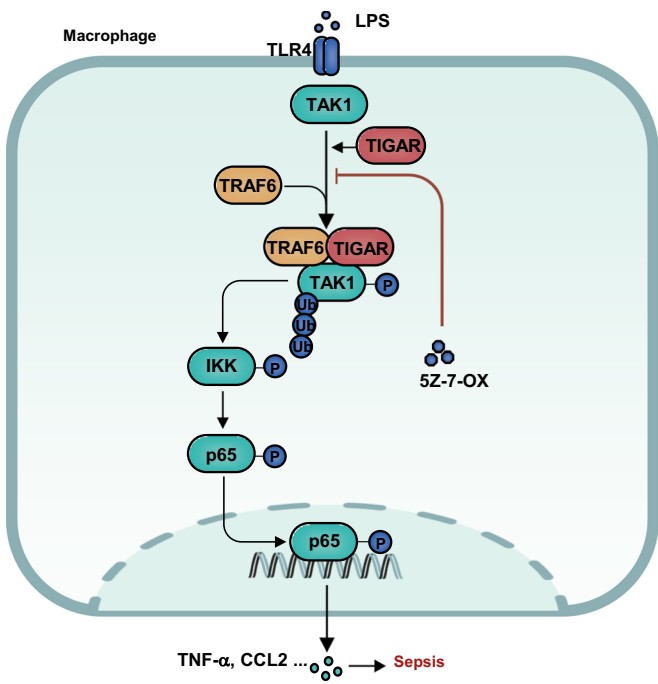

**Fig. 8 | Schematic diagram of the mechanism of TIGAR in sepsis.** Hypothetical mechanism model underlying macrophage TIGAR mediating inflammation in the context of sepsis.

confirmed the expression of the transfected Flag-TIGARs in lung stroma (Fig. S11e). The expression of TIGAR or TMU but not Mut2 impaired health conditions (Fig. 7b), decreased survival rates (Fig. 7c), exacerbated lung injury (Fig. S12a), and promoted the production of inflammatory cytokines in *Tigar* KO mice (Fig. 7d–f, Fig. S12b). These data clearly demonstrate the necessity of TIGAR structure instead of its phosphatase activity in promoting macrophage inflammation.

### Pharmacological inhibition of TIGAR binding to TAK1 antagonizes sepsis in mice

Discovery of the role of the TIGAR-induced TAK1 ubiquitination in activating NF-κB signaling propelled us to further explore its potential in reversing the disease process. Theoretically, 5Z-7-OX, an orthosteric covalent inhibitor of TAK1[26], has an ability to compete with residues 152–161 of TIGAR for binding the ATP-binding pocket of TAK1 (Fig. 7g). We verified it by performing experiments and found that 5Z-7-OX inhibited TIGAR-TAK1 binding in a concentration-dependent manner (Fig. 7h). It also prevented the TIGAR-induced TRAF6-TAK1 complex formation (Fig. 7i), suggesting that pharmacological interference may be useful to inhibiting TIGAR-facilitated inflammation.

We further examined the effect of 5Z-7-OX on inflammation in vivo. When 5Z-7-OX was intraperitoneally administered to CLP septic mice, it significantly improved the health conditions (Fig. 7j) and reduced levels of pro-inflammatory mediators in the body (Fig. 7k–n). The LPS-induced septic mice exhibited consistent phenotypic changes in the conditions of 5Z-7-OX treatment (Fig. S12c–g). The inhibitory efficacy of 5Z-7-OX on phosphorylation of TAK1 and p65 in lung tissues was confirmed at the end of the experiments (Fig. S12h). These data indicate that blocking the TIGAR binding to TAK1 and inhibiting TAK1 activation may inhibit inflammation, thereby attenuating sepsis progression.

## Discussion

Macrophages have functional plasticity implicated in diverse areas of biology including tissue development and maintenance, innate immunity and inflammation, and tissue repair[4,32]. The definite phenotype of macrophages is largely determined by the microenvironment where they reside. In response to stressing signals, the shifted functions of macrophages mounted by metabolic reprogramming contribute to the onset and progression of the relevant diseases[33]. However, the link between metabolic reprogramming and altered cellular activities is intricated in which metabolic enzymes and their products play pivotal roles[10,12]. Especially, the spatiotemporal dynamics in metabolic enzymes confer the enzymes multiple distinct functions including canonical functions in certain pathways as well as noncanonical, or non-metabolic, functions[12]. Deep insights into the functional switch in metabolic enzymes may provide a powerful interface for understanding macrophage heterogeneity. In this study, we reveal the function of TIGAR in the mediation of inflammatory IKK-NF-κB signaling that is independent of its canonical metabolic enzyme in macrophages. TIGAR elicits a complex formation with TAK1 and TRAF6 through protein–protein interaction and activation of TAK1, thereby promoting inflammation and accelerating the progression of sepsis (Fig. 8).

As a phosphatase, TIGAR hydrolyzes Fru-2,6-P2 to inhibit glycolysis and increases the cellular NADPH level to reduce intracellular ROS[14]. Its role in inflammation is still obscure. TIGAR has been reported to bind to LUBAC inhibiting NF-κB signaling pathway in adipocytes[23]. A similar effect of TIGAR has been documented in ischemic astrocytes[34]. During kainic acid-induced neuroinflammation, TIGAR attenuates the production of inflammatory mediators and NLRP3 pathway activation[35]. However, it has been found that TIGAR has no impact on NF-κB activation in the neocarzinostatin-treated glioma cells[36]. Furthermore, TIGAR even triggers autophagy to upregulate the expression of VEGF, thereby exacerbating chronic pulmonary inflammation[37]. Knockdown of TIGAR inhibits translocation of p65 into the nucleus in tumor cells[38]. The versatile properties of TIGAR in inflammatory response may be partly due to varying cell types and niches in which it appears. As a main immune cell in innate immunity, macrophage is thought as a pivotal modulator of inflammation in the body[4,32]. Macrophage responds to stress by activation of the transcriptional factors including p53, cAMP response element binding protein (CREB), and HIF1α, which are pro-inflammatory in nature[39–42]. Coincidently, these transcriptional factors can also bind to the promoter of TIGAR[21,43,44], explaining the high expression of macrophage TIGAR in septic mice and suggesting the role of TIGAR in macrophage inflammatory response. Indeed, TIGAR exhibited a stimulatory effect on IKK-NFκB signaling either under a short period of LPS agitation in vitro or under persistent stress conditions in vivo, in which TIGAR was up-regulated. These discoveries reveal that TIGAR activates IKK-NFκB signaling at both basal and up-regulated states. P53 and so on stress response molecules may indirectly promote inflammation via TIGAR-mediated IKK-NFκB signaling activation and lead to forming a vicious circle of inflammatory response in macrophages. The facilitative effect of TIGAR on inflammation is independent of its phosphatase activity as evident by mutation of the catalytic residues of TIGAR. Consistently, the phosphatase-free mechanism may underly a stimulative effect of TIGAR on ROS generation in macrophages (Fig. S4a, b). Thus, the metabolic enzyme TIGAR in macrophages may exert its noncanonical functions in response to those stressors inducing TIGAR upregulation, thereby contributing to inflammatory diseases.

It is known that the TAK1 kinase complex is recruited by ubiquitination of RIPK1 through the interaction between its polyubiquitin chains and TAB2. The polyubiquitin chains on RIPK1 also bind to NEMO, resulting in the recruitment of the IKK complex. IKKβ in the complex is then phosphorylated and activated by TAK1, leading to the phosphorylation, ubiquitination, and degradation of IκB. NF-κB subsequently translocates into the nucleus to activate gene expression[45,46]. TIGAR can interact with HK2, AKT, NRF2, and so on in different cells[15–18,23]. Our discovery reveals that TIGAR can simultaneously bind with TAK1 as well as E3 ligase TRAF6, instead of other endogenous

enzymes such as TAB1, USP4, and USP18, to form a complex in macrophages. Formation of TIGAR, TAK1, and TRAF6 complex leads to K63-linked ubiquitination of TAK1 and subsequent TAK1 autophosphorylation[47]. Although the TIGAR-regulated specific ubiquitination site of TAK1 warrants further investigation, our data suggest that this process should not involve TAB2/3, the known enhancer for TAK1 activity by linking TRAF6 to TAK1[29,30]. The TIGAR-induced TAK1 activation further induces IKK-NF-κB signaling activation.

As a key signaling node in the IKK-NF-κB pathway, TAK1 contributes to diverse inflammatory diseases including obesity-associated hepatic steatosis[48], inflammatory bowel disease[49], and cancer[50]. The application of TAK1 inhibitors as a potential therapeutic target has aroused great interest. For example, Takinib, an inhibitor of TAK1, has an up-regulative effect on programmed cell death targeting metastatic breast cancers and rheumatoid arthritis[51]. Another TAK1 inhibitor 5Z-7-OX can suppress both the kinase and adenosine triphosphatase activity of TAK1 and is widely used for biomedical research[26]. 5Z-7-OX increases the efficacy of inflammation inhibition in rheumatoid arthritis cell models[52]. However, as a nonselective natural product, 5Z-7-OX also effectively inhibits a panel of at least 50 other kinases and forms a covalent bond with reactive cysteines in the activation loop of its targets, producing several undesired side effects[26]. The therapeutic effects of 5Z-7-OX on sepsis may be attributed to its anti-inflammatory properties, which is rooted in preventing the interaction between TIGAR and TAK1. Surely more specific and stronger TAK1 inhibitors need to be developed to interfere with the TIGAR-TAK1 interactive interface. Obviously, our findings pave a promising avenue to block TIGAR-TAK1 interaction by a biophysical mechanism instead of a classical enzyme-catalyzed reaction to avoid possible off-target effects.

## Methods

### Mice
Animal protocols were reviewed and approved by the Animal Care and Use Committee of Nanjing Medical University. All mice used in this study were bred and maintained in the animal facility of the Nanjing Medical University on a 12 h light-dark cycle at a room temperature 22–24 °C and humidity 30–70% with free access to food and water. *Tigar* KO mice were kindly provided by Dr. Yaoyu Chen (Nanjing Medical University, China), and *Tigar*^flox/flox mice were provided by Dr. Zhenghong Qin (Soochow University, China). To generate the myeloid *Tigar*-deficient mice, *Tigar*^flox/flox mice were crossed with *Lyz2*-Cre mice (B6.129P2-*Lyz2*^tm1(cre)Ifo/J, *Lyz2*-Cre^KI/KI). Two sets of primers were used to genotype the mice by PCR on genomic DNA isolated from tails (*Tigar*-F1 5-AGCTTCCTCCACAGTGCTGAGATC and *Tigar*-R1 5-TGATGTTC-CAAAGGAGACAGTAAA; *Tigar*-F2 5-GACATTTACAG GCTCGCATTAG-CAC and *Tigar*-R2 5-GTCAGGGTATGTGCA TTTGACTG). In this study, *Tigar*^flox/flox*Lyz2*-Cre^WT/WT mice were defined as the MacWT mice, and *Tigar*^flox/flox*Lyz2*-Cre^KI/KI mice were defined as MacKO mice.

### Animal models
Male mice were used in this study to avoid the possible influence of estrogen on survival and inflammation in septic mice[53]. LPS (L4516, Sigma-Aldrich, 10 mg kg⁻¹) was intraperitoneally administrated to induce endotoxemia. After 12 h 8-week-old male MacWT and MacKO mice were euthanized by CO₂ inhalation and sacrificed for determination of inflammatory factors. The 8-week-old male MacWT and MacKO murine survival rate was monitored during 96 h of LPS administration or CLP surgery. To construct the CLP septic model, 8-week-old male MacWT and MacKO male mice were anesthetized with 2% isoflurane inhalation. Abdominal hair was removed, and the skin was disinfected. A midline laparotomy was performed in a sterile field. The cecum was exposed and was ligated and punctured twice with a 19-gauge needle. This procedure was followed by slight compression to release a small amount of intestinal contents from

the holes. Then, the cecum was returned to the abdominal cavity, and the abdominal skin was closed. Next, mice were placed on a 37 °C heating plate until awakening. After 12 h mice were euthanized by CO₂ inhalation and sacrificed for determination of inflammatory factors[54–56]. For TAK1 inhibitor therapeutic experiments, the TAK1 inhibitor 5Z-7-OX (O9890, Sigma-Aldrich) was dissolved in 10% dimethyl sulfoxide and was intraperitoneally injected into 8-week-old C57BL/6J male mice at a dose of 5 mg kg⁻¹ starting 1 h before LPS challenge or CLP surgery.

### Establishment of TIGAR, TMU, and Mut2-re-expressing mice
We constructed the myeloid-specific CD11b promoter-driven lentiviral plasmids (pCDH-CD11b-T2A-copGFP) encoding wild-type TIGAR (Lenti-TIGAR), TIGAR catalytic dead mutant (Lenti-TMU), or TIGAR mutant lacking TAK1 binding motif (Lenti-Mut2), respectively[57–59]. Lentiviruses expressing TIGAR, TMU, or Mut2 in macrophages were injected into *Tigar* KO mice via tail vein. Four days later, the mice were performed CLP surgery.

### Clinical score
For clinical score, mice were assessed on 0, 6, and 12 h after LPS injection in a blinded fashion. At indicated time points, animals were assessed for symptom scores by grading the severity of conjunctivitis, diarrhea, ruffled fur, and lethargy on a three-point scale (0, 1, and 2). The means of the three assessments were used for grading. Conjunctivitis: score 0-eyes closed or bleared with serous discharge; score 1-eyes opened with serous discharge; score 2-normal, no conjunctivitis. Stool consistency: score 0-diarrhea; score 1-loose stool; score 2-normal stool. Hair coat: score 0-rough and dull fur, ungroomed; score 1-reduced grooming, rough hair coat; score 2-well groomed, shiny fur. Activity upon moderate stimulation: score 0-lethargic, only lifting of the head after moderate stimulation; score 1-inactive, less alert, <2 steps after moderate stimulation; score 2-normal locomotion and reaction, >2 steps after moderate stimulation[60,61]. The total maximal score is 8, which indicates a normal health condition.

### Histological analysis
For immunofluorescence analysis, anti-TIGAR (rabbit, ab189164,1:100, Abcam), anti-p65 (rabbit, 8242, 1:100, Cell Signaling Technology), and anti-TAK1 (mouse, sc-166562, 1:100, Santa Cruz) antibodies were applied at 4 °C overnight. The secondary antibodies including Alexa Fluor 555 donkey anti-mouse IgG (A-31570, Thermo Fisher Scientific), and Alexa Fluor 488 donkey anti-rabbit IgG (A-21206, Thermo Fisher Scientific) were applied to react with primary antibodies for 1.5 h at 37 °C. The sections were spin-dried and mounted with DAPI Fluoromount-G (0100-20, Southern Biotech). Pictures were captured using a Carl Zeiss microscope and analyzed with Image-Pro Plus software.

Lung tissues were fixed in 4% paraformaldehyde, embedded in paraffin, and sectioned at 5 μm thickness. For IHC staining, sections were incubated with Flag antibody (rabbit, 14793, 1:100, Cell Signaling Technology) followed by incubation with the secondary antibody conjugated with horseradish peroxidase. The sections were then treated with the ABC staining system (sc-2018, Santa Cruz) according to the instructions of the manufacturer. For all sections, 3,3-diaminobenzidine was used as the indicator substrate, which appeared as a brown reaction product. The histological features were observed and captured under a light microscope (Carl Zeiss).

Lung histopathological injury was assessed by hematoxylin and eosin (H&E) staining. Three randomly chosen fields of each section were scored according to alveolar congestion, hemorrhage, infiltration or aggregation of inflammatory cells in the airspace or vessel wall, and thickness of the alveolar wall/hyaline membrane formation. Each section was scored on a five-point scale as follows: 0 = minimal damage; 1 = mild damage; 2 = moderate damage; 3 = severe damage;

and 4 = maximal damage[62,63]. The histological features were observed and captured under a light microscope (Carl Zeiss).

## Flow cytometry analysis

Murine spleens were excised, minced, and filtered in PBS. Cell suspension was filtered through a 200 μm cell strainer and centrifuged at $900 \times g$ for 10 min at 4 °C. Whole blood was collected and red blood cells (RBC) were lysed using RBC lysis buffer. Cells were resuspended in PBS supplemented with 1% FBS and stained with indicated fluorescent isotope conjugated antibodies for 30 min at room temperature in the dark. The antibodies used for FACS included APC anti-CD11b (553312, 1:100, BD Pharmingen), PE anti-CD45 (553081, 1:100, BD Pharmingen). Fluorescence-activated cell analysis data were collected with a BD FACSymphony A5 SORP and analyzed with FlowJo v.10 (Fig. S13).

## Analysis of metabolic parameters

After a 6–8 h fasting, mice were anesthetized, and blood samples were collected in Heparin-coated tubes. After centrifugation at $900 \times g$ for 15 min at 4 °C, plasma was separated and stored at −80 °C. Plasma concentrations of mouse TNF-α (BMS6005, eBioscience), CCL2 (BMS6007-3, eBioscience), and CCL8 (CSB-E07458m, eLabscience) in these samples were quantified by sandwich enzyme-linked immunosorbent assay (ELISA) kits.

## Nitrite and ROS detection

Nitrite level is used to assess nitric oxide (NO) production. Culture supernatant was collected for measuring the production of NO using the Griess reagent (Sigma-Aldrich). For determination of ROS production, murine BMDMs were treated with LPS (100 ng ml⁻¹) for 8 h and then were treated with 10 mM 2′,7′-Dichlorodihydrofluorescein (DCFH-DA) (Beyotime) for 30 min. The fluorescence intensity of 2′,7′-dichlorofluorescein (DCF) was evaluated by flow cytometry.

## Kinase assay in vitro

TAK1 kinase activity was assayed using TAK1-TAB1 kinase enzyme system (V4089, Promega) and Universal Kinase Assay Kit (Ab138879, Abcam). Briefly, TAK1 kinase, ATP, and different doses of purified TIGAR protein (Ag17661, Protein Tech) were mixed in a kinase reaction buffer containing DTT for 30 min for completion of the kinase reaction. ADP sensor buffer and ADP sensor were added into each well and the reaction mixture was incubated in the dark at room temperature for 45 min. After the reaction, the fluorescence intensity was detected with a microplate fluorescence reader at Ex/Em = 540/590 nm.

## RNA-sequencing

Murine bone marrow-derived macrophages (BMDMs) treated by LPS (100 ng ml⁻¹) for 8 h were subjected to RNA-sequencing analysis. Total RNA samples meeting the following requirements were used in subsequent experiments: RNA integrity number (RIN) > 7.0 and a 28S:18S ratio > 1.8.RNA-seq libraries were generated and high-throughput RNA sequencing was performed by Illumina Hi Seq 2500 (Illumina, San Diego, CA, USA) at Capital Bio Corporation (Beijing, China). Raw RNA-sequencing data were quality controlled with Fast QC (v0.11.5) and then low-quality data were filtered using NGSQC (v2.3.3). The clean reads were mapped to the mouse genome (GRCm38.p5) by HISAT2. Gene expression analyses were performed with String Tie (v1.3.3b). DESeq (v1.28.0) was used to identify differentially expressed genes. Parameters for classifying significantly DEGs are ≥2 fold differences (|log2FC| ≥ 1, FC: the fold change of expressions) in the transcript abundance and $p \le 0.05$. Gene ontology and KEGG enrichment analysis were conducted using KOBAS. Raw data files and processed files have been deposited in the gene expression omnibus under accession no. GSE202446.

## Cell culture

HEK293 and RAW264.7 cells were obtained from ATCC. HEK293 cells were maintained in Dulbecco's modified Eagle's medium (DMEM)/F12 medium supplemented with 10% fetal bovine serum (FBS), and 1% penicillin-streptomycin (PS). RAW264.7 cells were cultured in DMEM supplemented with 10% FBS, and 1% PS. Mouse PMs and BMDMs were prepared as follows. For the isolation and culture of PMs, the mouse abdomen was sterilized using 75% ethanol. The abdominal cavity was shaken with sterile PBS, and then the peritoneal fluid containing the macrophages were collected, pooled, and centrifuged. Cell pellets were suspended in DMEM containing 10% FBS and 1% PS. After 2 h incubation at 37 °C, non-adherent cells were removed by washing, and the adherent cells were maintained for 24 h in a 10% serum-containing medium for further study. BMDMs were flushed from the hindlimbs of mice. Cells were maintained for 7 days in RPMI 1640 medium supplemented with 10 ng ml⁻¹ mouse macrophage colony stimulation factor (mMCSF, 416 ML, R&D), 10% FBS, and 1% PS and allowed to differentiate into mature macrophages. The medium was replenished on days 3 and 5[64]. Cells were incubated with 100 ng ml⁻¹ LPS at indicated times for analysis.

## Spleen and blood F4/80⁺ cells separation

For isolation of F4/80⁺ cells, murine spleens were excised, minced, and filtered in PBS. The cell suspension was filtered through a 200 μm cell strainer and centrifugated at $900 \times g$ for 10 min at 4 °C. Red blood cells in cell suspension were lysed with red blood cell lysis buffer for 5 min. The homogeneous cell suspensions were resuspended in PBS supplemented with 3% FBS. After intensive washing, F4/80⁺ cells were purified using magnetic beads (Invitrogen) according to the manufacturer's instructions. Cells were immediately used for total protein and RNA extraction. For the preparation of murine peripheral blood F4/80⁺ cells, peripheral blood was diluted with PBS, followed up by steps mentioned above.

## Co-immunoprecipitation (Co-IP)

Cells were lysed in cell lysis buffer (P0013, Beyotime, China) containing protease inhibitor cocktail tablets (04693132001, Roche, Germany) on ice for 30 min. Supernatants were collected following centrifugation at $13,800 \times g$ at 4 °C for 15 min. After centrifugation, the cell lysates were incubated with the indicated antibody with gentle rocking at 4 °C overnight, and protein A/G plus agarose beads (sc-2003, Santa Cruz) were added for 4–6 h at 4 °C. The sepharose samples were collected by centrifugation, washed three times with cell lysis buffer, and the immunoprecipitants were eluted with sample loading buffer by boiling. After SDS-PAGE, equivalent amounts of proteins were electroblotted onto polyvinylidene fluoride (PVDF) membranes with the appropriate antibodies. Antibodies used for Co-IP were anti-TAK1 (5206, Cell Signaling Technology), anti-TRAF6 (PA5-29622, Thermo Fisher Scientific), anti-Flag (F1804, Sigma-Aldrich), anti-HA (26183, Thermo Fisher Scientific), and anti-His (MA1-21315, Thermo Fisher Scientific).

## Western blot analysis

Whole-cell lysates or tissue lysates were extracted from tissues or cells in protein lysis buffer supplemented with a complete protease inhibitor cocktail and phosphatase inhibitors (0490684001 Roche). Protein concentration was measured by bicinchoninic acid assay (BCA) assay (23225, Thermo Fisher Scientific, UK) according to the manufacturer's instructions. Proteins were boiled for 5 min and separated on 8–10% polyacrylamide gel. Equal concentrations of protein were resolved on SDS-PAGE and transferred to PVDF membranes. The membranes were incubated with either 5% non-fat milk or 5% BSA for 2 h and then probed overnight at 4 °C with primary antibodies. Antibodies were against TIGAR (sc-166290, 1:500, Santa Cruz), TAK1 (sc-166562, 1:500, Santa Cruz), phospho-TAK1 (MA5-15073, 1:1000,

Thermo Fisher Scientific), phospho-IKKα/β (2697, 1:1000, Cell Signaling Technology), IKKα/β (sc-7607, 1:500, Santa Cruz), IκBα (4814, 1:1000, Cell Signaling), phospho-p65 (3033, 1:1000, Cell Signaling Technology), p65 (8242, 1:1000, Cell Signaling Technology), HOIP (RNF31) (A303-560A, 1:500, Bethyl), SHARPIN (ab197853, 1:1000, Abcam), HOIL-1 (RBCK1) (sc-365523,1:200, Santa Cruz), Flag (F1804, 1:1000, Sigma-Aldrich), HA (11867423001, 1:1000, Roche), HA (26183, 1:5000, Thermo Fisher Scientific), Ub (MAB1510, 1:1000, Millipore), K63-Ub (5621, 1:1000, Cell Signaling Technology), His (66005-1-Ig, 1:1000, Protein Tech), Myc (2278, 1:1000, Cell Signaling Technology), His (MA1-21315, 1:1000, Thermo Fisher Scientific), α-Tubulin (11224-1-AP, 1:1000, Protein Tech), GAPDH (KC-5G4, 1:3000, Kangchen Tech), β-actin (sc-47778, 1:1000, Santa Cruz), Lamin B1 (66095-1-Ig, 1:1000, Protein Tech), followed by incubation with the corresponding secondary antibodies horseradish peroxidase-conjugated secondary antibody for 2 h at room temperature. After washing with TBST three times for 10 min each, the protein bands were visualized using super signal maximum sensitivity substrate (34096, Therma Fisher Scientific) and imaged by Chemi Doc XRS⁺ imaging system (Bio-Rad) and quantified with the ImageJ. Nuclear fractions were separated using nuclear and cytoplasmic extraction reagents kit (78833, Thermo Fisher Scientific).

### RT-qPCR analysis
Total RNA from tissues and purified cells was extracted using RNAiso Plus (TaKaRa, Japan) and reverse-transcribed using commercial kits (Vazyme Biotech, China). The quality and concentration of RNA were determined by absorbance at 260 and 280 nm using a NanoDrop spectrophotometer. RT-qPCR was performed using AceQ qPCR SYBR Green Master Mix (Q131-02, Vazyme Biotech, China) and an ABI QuantStudio 6 system (Applied Biosystems). The data were normalized by glyceraldehyde-3-phosphate dehydrogenase (GAPDH). The primer sequence for RT-qPCR used in this study is provided in Supplementary Table 1.

### Protein–protein docking
The structure of TAK1 (26-303) was obtained from the Protein Data Bank (PDB code: 7NTH), with all missing loops filled in by the SWISS-MODEL server[65]. The structure of TIGAR was predicted by AlphaFold2[66]. Prior to docking, all proteins were optimized using the default parameters of the protein preparation workflow in Maestro (Schrödinger, Inc). The protein–protein docking was then performed using the default parameters of ClusPro[67]. The docking results were analyzed using default parameters of dimer classification[68] and Prime MM-GBSA (Schrödinger, Inc) calculations, combined with visual inspection, to determine an initial binding conformation.

### Molecular dynamics (MD) simulations
MD simulations were performed using Desmond[69] package of Schrödinger 2022-2 with the DES-Amber force field[70]. The binding model of TAK1-TIGAR obtained in the last step was explicitly solvated with TIP3P water molecules[71] under cubic periodic boundary conditions for a 15 Å buffer region. The overlapping water molecules were deleted and 0.15 M KCl was added. The systems were neutralized by adding K⁺ as counter ions. Brownian motion simulation was used to relax these systems into local energy minimum states separately. An ensemble (NPT) was then applied to maintain the constant temperature (310 K) and pressure (1.01325 bar) of the systems, and the simulations were started with different random initial velocities in triplicate at the temperature of 310 K. To avoid artifacts caused by the periodicity, we processed the trajectory using the *trj_center.py* (Schrödinger, Inc). Root mean square deviation (RMSD) was calculated based on C-alpha and plotted using OriginPro2022b. The *trj_essential_dynamics.py* (Schrödinger, Inc.) was used to calculate the eigenvectors and eigenvalues. Eigenvectors (or principal components) represent positional deviations, and the magnitude of atomic fluctuations is associated with eigenvalues. Per-frame conformational deviations are projected onto the calculated modes (principal component space). Gmx sham and Matplotlib were used to calculate and plot the three-dimensional Gibbs free energy surface with PC1 and PC2. Additionally, the three-dimensional structures of proteins and small molecules were visualized using ChimeraX[72].

### Statistical analysis
Statistical analyses were performed using GraphPad Prism 8 software and results are represented as mean ± SEM. Data normality was examined by Shapiro–Wilk test ($n < 10$). Data homoscedasticity was assessed with the Bartlett and Brown–Forsythe test. For normally distributed data, comparisons between 2 groups were performed using Student $t$-test, and comparisons among multiple groups were performed using ANOVA followed by Bonferroni's multiple comparisons test. For non-normal data comparisons were performed by the Mann–Whitney U test or the Kruskal–Wallis test followed by Dunn's multiple comparison test. $P < 0.05$ were considered statistically significant.

### Reporting summary
Further information on research design is available in the Nature Portfolio Reporting Summary linked to this article.

## Data availability
The raw bulk RNA-sequencing data generated in this study have been deposited in the gene expression omnibus under accession no. GSE202446. All data associated with this study are in the paper and the Supplementary Materials. Source data are provided with this paper.

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

## Acknowledgements

Thanks to Yaoyu Chen and Zhenghong Qin for providing us the *Tigar* KO and *Tigar*^flox/flox mice respectively. pCDH-CD11b-T2A-copGFP plasmid was gifted by Guo Zhang. This work was supported by grants from the National Natural Science Foundation of China (81830011, 82030012 to Q.C., 82270476, 82070457 to J.B., 82300515 to D.W., 82100433 to B.J., 82170444 to X.Z., 82270361 to H.Z.), the National Key R&D Program of China (2022YFC3400501 to F.B.), the Natural Science Foundation of the Jiangsu Higher Education Institutions of China (20KJA310007 to X.Z.) and Jiangsu Zhuoyue Postdoctoral Project (2023ZB029 to D.W.).

## Author contributions

J. Ben and Q. Chen conceived and designed the experiments. D. Wang, Y. Li, X. Shen, X. Shi, C. Li, and Y. Zhang performed the experiments. D. Wang, Y. Li, and Y. Zhang analyzed the data. Y. Ji provided intellectual advice. H. Yang, F. Bai, X. Liu, and W. Gao did the computational modeling, and structural analysis and wrote the part. B. Jiang, X. Zhu, H. Zhang, X. Li, H. Bai, and Q. Yang provided technical assistance. D. Wang, H. Yang, J. Ben, and Q. Chen wrote the paper. All authors read and approved the final manuscript.

## Competing interests

The authors declare no competing interests.
