## [Peer Review File · Nature Communications]

Disruption of TIGAR-TAK1 alleviates immunopathology in a murine model of sepsisREVIEWER COMMENTS

Reviewer #1 (Remarks to the Author):

The authors report TIGAR's new binding partner, TAK1, which they propose as a target of TIGAR regulation of NF- κ B pathway. The finding is novel and potentially important in understanding of TIGAR biology. However, many shortcomings dampen rigor of this study as listed below.

Major points:

1) TAK1 inhibitor 5z-7 oxozeaenol was solely used to test the role of TIGAR regulation of TAK1 in the in vivo studies (Fig. 8). This experimental design has a fundamental flaw. 5z-7 oxozeaenol inhibits TAK1 catalytic activity independently of TIGAR presence. Thus, the Fig. 8 results only indicate TAK1 inhibition suppress inflammation but not suggest any involvement of TIGAR. TIGAR mutant lacking only TAK1 binding may be utilized to determine the importance of TIGAR regulation of TAK1-inflammation.

2) Lys2-Cre system is known to be leaky. It should be more convincing if the phenotypes seen in Fig. 1-3 would be confirmed by using other deletion systems. Alternatively, TIGAR catalytic dead mutant knock-in mice would be highly informative.

3) Interaction partners of TIGAR was identified by testing only several NF- κ B pathway molecules, which is not an ideal method. Performing an unbiased screening would be more rigorous. While LUBAC is known to interact with TIGAR, it was not tested in this study.

4) 5z-7 oxozeaenol is a selective but not specific inhibitor of TAK1 as the authors discussed. Alternative TAK1 inhibitors, such as Takinib, should also be tested.

5) While TIGAR was upregulated at 12-24 hours after LPS stimulation, the effect of Tigar deletion was tested at 15-30 min. It is not well discussed whether basally expressed and/or induce TIGAR is important in NF- κ B regulation. Whether and how p53 is involved in this pathway is not discussed.

6) The role of TAK1 polyubiquitination has not been well defined. RIPK1 and NEMO anchored poly-ubiquitin chains are known to play major roles in NF- κ B signaling. More rigorous discussion is required.

7) Fig. 8 shows TIGAR binds to the ATP binding pocket of TAK1. It seems that TIGAR structurally interferes with substrate binding of TAK1. Testing whether TAK1-TIGAR binding upregulates TAK1 catalytic activity in vitro using purified proteins would be needed.

Reviewer #2 (Remarks to the Author):

The authors have found that mouse TIGAR-TAK1 are involved in the multiorgan dysfunction developing in mice with sepsis or atherosclerosis. By disrupting TIGAR-TAK1, this has caused alleviation of mouse sepsis and atherosclerosis due to "suppressed inflammatory responses." Using mouse macrophages that are proinflammatory, they found that blockade of TP53 reduced glycolysis and apoptosis, reducing effects of sepsis and the inflammatory lesions developing in atherosclerosis. Such infirmities were greatly reduced by the

manipulations involving TIGAR. It is clear that both disorders involve a large number of inflammatory systems such as the clotting system in both disorders. While the in vitro data seem relatively straight-forward, the design of the studies does not allow the reader to assume that important resulting events were related to a role for macrophage products and the accompanying results. Such concern also involves whether the in vivo events can be linked to macrophage products. Such limitations restrict many of the claims of the authors and do not indicate if important signaling pathways have been blocked, which could lead to serious blockade of the innate immune system.

Response to reviewers' comments

Reviewer #1 (Remarks to the Author):

The authors report TIGAR's new binding partner, TAK1, which they propose as a target of TIGAR regulation of NF- κ B pathway. The finding is novel and potentially important in understanding of TIGAR biology. However, many shortcomings dampen rigorousness of this study as listed below.

Major points:

1) TAK1 inhibitor 5z-7 oxozeaenol was solely used to test the role of TIGAR regulation of TAK1 in the in vivo studies (Fig. 8). This experimental design has a fundamental flaw. 5z-7 oxozeaenol inhibits TAK1 catalytic activity independently of TIGAR presence. Thus, the Fig. 8 results only indicate TAK1 inhibition suppress inflammation but not suggest any involvement of TIGAR. TIGAR mutant lacking only TAK1 binding may be utilized to determine the importance of TIGAR regulation of TAK1-inflammation.

A: Thank the reviewer for raising this important issue and providing us valuable suggestions.

According to the reviewer's suggestion, we used lentiviral plasmids of wild type TIGAR (Lenti-TIGAR), TIGAR catalytic dead mutant (Lenti-TMU), or TIGAR mutant lacking TAK1 binding motif (Lenti-Mut2) driven by myeloid specific CD11b promoter¹⁻³ to infect *Tigar* KO mice, respectively. Our results showed that re-expression of Mut2 in macrophages did not promote inflammation and exacerbate cecal ligation puncture (CLP) sepsis in *Tigar* KO mice, which are different from those of Lenti-TIGAR or Lenti-TMU (Fig 7a-f, Fig. S11a). These data indicate that TIGAR promote

macrophage inflammation dependent on its binding with TAK1.

To improve the sepsis-related research work, we newly generated CLP-induced septic model, a more relevant physiological sepsis mouse model⁴⁻⁶, to investigate the role of macrophage TIGAR. The results in murine CLP-induced sepsis confirmed the phenotypic changes in LPS-inoculated mice (Fig. 1k-n). 5Z-7-OX also exhibited an improvement effect on the health conditions (Fig.7j) and reduced levels of pro-inflammatory mediators (Fig.7k-n) in CLP septic mice. Our new data from CLP model strengthen the discovery from the LPS-induced endotoxemia. These new data have been included in the revised manuscript. Correspondingly, the data on atherosclerosis have been deleted in the revised manuscript.

2) Lys2-Cre system is known to be leaky. It should be more convincing if the phenotypes seen in Fig. 1-3 would be confirmed by using other deletion systems. Alternatively, TIGAR catalytic dead mutant knock-in mice would be highly informative.

A: Thanks for the reviewer's kind suggestion. We newly performed experiments per the reviewer's indication. The results have been mentioned above (Fig 7a-f, Fig. S11a). Re-expression of TIGAR catalytic dead mutant (Lenti-TMU) in macrophages exacerbated inflammation and CLP sepsis in *Tigar* KO mice as effective as TIGAR.

3) Interaction partners of TIGAR was identified by testing only several NF-kB pathway molecules, which is not an ideal method. Performing an unbiased screening would be more rigorous. While LUBAC is known to interact with TIGAR, it was not tested in this

study.

A: Yes, we agree with the reviewer. We newly examined the interaction of TIGAR with the subunits HOIP, HOIL-1, and SHARPIN of linear ubiquitination assembly complex (LUBAC) according to this reviewer's kind suggestion. The results showed that none of LUBAC subunits interacted with TIGAR in macrophages (Fig. S6a), though HOIP has been previously reported to bind to TIGAR in adipocytes⁷.

4) 5z-7 oxozeaenol is a selective but not specific inhibitor of TAK1 as the authors discussed. Alternative TAK1 inhibitors, such as Takinib, should also be tested.

A: Per the reviewer's suggestion, we used an alternative TAK1 inhibitor Takinib to treat macrophages. We found that Takinib had consistent effect with 5Z-7-OX on TIGAR-induced inflammation in macrophages (Fig. S6d).

5) While TIGAR was upregulated at 12-24 hours after LPS stimulation, the effect of Tigar deletion was tested at 15-30 min. It is not well discussed whether basally expressed and/or induce TIGAR is important in NF- κ B regulation. Whether and how p53 is involved in this pathway is not discussed.

A: According to the reviewer's suggestion, we expanded the relevant part of discussion as follows.

“Macrophage responds to stress by activation of the transcriptional factors including p53, cAMP response element binding protein (CREB), and HIF1 α , which are proinflammatory in nature⁸⁻¹¹. Coincidentally, these transcriptional factors can also bind

to the promoter of TIGAR, explaining the high expression of macrophage TIGAR in septic mice and suggesting the role of TIGAR in macrophage inflammatory response¹²⁻¹⁴. Indeed, TIGAR exhibited stimulatory effect on IKK-NFκB signaling either under short period of LPS agitation *in vitro* or under the persistent stress conditions *in vivo*, in which TIGAR was up-regulated. These discoveries reveal that TIGAR activates IKK-NFκB signaling at both basal and up-regulated states. p53 and so on stress response molecules may indirectly promote inflammation via TIGAR-mediated IKK-NFκB signaling activation and lead to forming a vicious circle of inflammatory response in macrophages.” in lines 332-343.

6) The role of TAK1 polyubiquitination has not been well defined. RIPK1 and NEMO anchored poly-ubiquitin chains are known to play major roles in NF-κB signaling. More rigorous discussion is required.

A: Per the reviewer’s suggestion, we re-edited the relevant parts of discussion as follows.

“Formation of TIGAR, TAK1, and TRAF6 complex leads to K63-linked ubiquitination of TAK1 and subsequent TAK1 autophosphorylation. Ubiquitylation of RIPK1 provides multiple Ub scaffolds for the recruitment of TAB1/2 and activates TAK1¹⁵.” in lines 353-356.

“Ubiquitination of NEMO prevents its binding with IKKα/β, leading to the activation of canonical IKK-NF-κB signaling¹⁶.” In lines 360-361.

7) Fig. 8 shows TIGAR binds to the ATP binding pocket of TAK1. It seems that TIGAR structurally interferes with substrate binding of TAK1. Testing whether TAK1-TIGAR binding upregulates TAK1 catalytic activity *in vitro* using purified proteins would be needed.

A: We agree at the important issue addressed by the reviewer. Per the reviewer's suggestion, we tested whether TAK1-TIGAR binding influence TAK1 catalytic activity *in vitro* using TAK1-TAB1 kinase enzyme system (V4089, Promega) and Universal Kinase Assay Kit (Ab138879, Abcam). The results showed that TAK1 kinase catalytic activity was not directly affected by the presence of TIGAR *in vitro* (Fig. S6b).

Reviewer #2 (Remarks to the Author):

*The authors have found that mouse TIGAR-TAK1 are involved in the multiorgan dysfunction developing in mice with sepsis or atherosclerosis. By disrupting TIGAR-TAK1, this has caused alleviation of mouse sepsis and atherosclerosis due to "suppressed inflammatory responses." Using mouse macrophages that are proinflammatory, they found that blockade of TP53 reduced glycolysis and apoptosis, reducing effects of sepsis and the inflammatory lesions developing in atherosclerosis. Such infirmities were greatly reduced by the manipulations involving TIGAR. It is clear that both disorders involve a large number of inflammatory systems such as the clotting system in both disorders. While the *in vitro* data seem relatively straight-forward, the design of the studies does not allow the reader to assume that important resulting events were related to a role for macrophage products and the accompanying results. Such*

concern also involves whether the in vivo events can be linked to macrophage products. Such limitations restrict many of the claims of the authors and do not indicate if important signaling pathways have been blocked, which could lead to serious blockade of the innate immune system.

A: Thank the reviewer for the valuable comments and suggestions. Per the reviewer guidance we re-edited the manuscript focusing on septic models and deleted the data on atherosclerosis. To improve the sepsis-related research work, we performed new experiments in cecal ligation puncture (CLP)-induced septic model, a more relevant physiological sepsis mouse model⁴⁻⁶, to investigate the role of macrophage TIGAR. The results in murine CLP-induced sepsis confirmed the phenotypic changes in LPS-inoculated mice (Fig. 1k-n), which would be irrelevant to the clotting system since no changes in tail bleeding time and activated partial thromboplastin time (APTT) by ablation of myeloid *Tigar* (Fig. S1b-c). Our new data from CLP model strengthen the discovery from the endotoxemia and have been included in the revised manuscript.

We newly constructed wild type TIGAR (Lenti-TIGAR), TIGAR catalytic dead mutant (Lenti-TMU), and TIGAR mutant lacking TAK1 binding motif (Lenti-Mut2) lentiviral plasmids directed by myeloid specific CD11b promoter¹⁻³ to infect *Tigar* KO septic mice, respectively. We found that re-expression of both TIGAR and TMU but not Mut2 in macrophages impaired health conditions and promoted inflammation in mice (Fig 7a-f, Fig. S11a). These results further confirm our discovery that macrophage TIGAR exerts a non-canonical function to promote inflammation in sepsis.

References

1. Dziennis, S., *et al.* The CD11b promoter directs high-level expression of reporter genes in macrophages in transgenic mice. *Blood* **85**, 319-329 (1995).
2. Moreno-Lanceta, A., *et al.* RNF41 orchestrates macrophage-driven fibrosis resolution and hepatic regeneration. *Sci Transl Med* **15**, eabq6225 (2023).
3. Pahl, H.L., Rosmarin, A.G. & Tenen, D.G. Characterization of the myeloid-specific CD11b promoter. *Blood* **79**, 865-870 (1992).
4. Yan, Z., *et al.* Targeting adaptor protein SLP76 of RAGE as a therapeutic approach for lethal sepsis. *Nat Commun* **12**, 308 (2021).
5. Xu, L., *et al.* IL-33 induces thymic involution-associated naive T cell aging and impairs host control of severe infection. *Nat Commun* **13**, 6881 (2022).
6. Toscano, M.G., Ganea, D. & Gamero, A.M. Cecal ligation puncture procedure. *J Vis Exp* **51**, 2860 (2011).
7. Tang, Y., *et al.* The fructose-2,6-bisphosphatase TIGAR suppresses NF-kappaB signaling by directly inhibiting the linear ubiquitin assembly complex LUBAC. *J Biol Chem* **293**, 7578-7591 (2018).
8. Yoshida, Y., *et al.* p53-Induced inflammation exacerbates cardiac dysfunction during pressure overload. *J Mol Cell Cardiol* **85**, 183-198 (2015).
9. Spehlmann, M.E., *et al.* Trp53 deficiency protects against acute intestinal inflammation. *J Immunol* **191**, 837-847 (2013).
10. Kotla, S., Singh, N.K., Heckle, M.R., Tigyi, G.J. & Rao, G.N. The transcription factor CREB enhances interleukin-17A production and inflammation in a

mouse model of atherosclerosis. *Sci Signal* **6**, ra83 (2013).

11. Tannahill, G.M., *et al.* Succinate is an inflammatory signal that induces IL-1beta through HIF-1alpha. *Nature* **496**, 238-242 (2013).
12. Rajendran, R., *et al.* Acetylation mediated by the p300/CBP-associated factor determines cellular energy metabolic pathways in cancer. *Int J Oncol* **42**, 1961-1972 (2013).
13. Zou, S., *et al.* CREB, another culprit for TIGAR promoter activity and expression. *Biochem Biophys Res Commun* **439**, 481-486 (2013).
14. Bensaad, K., *et al.* TIGAR, a p53-inducible regulator of glycolysis and apoptosis. *Cell* **126**, 107-120 (2006).
15. Clucas, J. & Meier, P. Roles of RIPK1 as a stress sentinel coordinating cell survival and immunogenic cell death. *Nat Rev Mol Cell Biol* **24**, 835-852 (2023).
16. Clark, K., Nanda, S. & Cohen, P. Molecular control of the NEMO family of ubiquitin-binding proteins. *Nat Rev Mol Cell Biol* **14**, 673-685 (2013).

REVIEWER COMMENTS

Reviewer #1 (Remarks to the Author):

The authors have added several new results in this revised version. The major addition is the in vivo lentivirus transduction to Tigar-deficient mice. This new result showed that wild type and the enzyme dead mutant but not the TAK1 binding lacking mutant restored CLP-induced inflammation in Tigar-deficient mice.

The additional results are intended to be sufficient to address the reviewers' concerns in physiological significance of the finding. However, the new results are not convincing due to missing critical controls and supporting data (see the major points below).

Overall, the revised manuscript does not sufficiently demonstrate the importance of the macrophage-derived metabolism-independent TIGAR regulation of inflammation in the in vivo setting.

Major points:

- 1) The new results using myeloid specific CD11b promoter-driven lentivirus of Flag-TIGAR, Flag-TMU or Flag-Mut2 should be very significant additions. However, the results are not convincing. It is known that myeloids are difficult cell types in lentivirus infection. The data showing the efficiency of transduction should have been included. What proportion of circulating and tissue myeloids express the transduced proteins? It should also have been confirmed whether other cell types did express the transduced TIGAR.
- 2) The authors should present the mouse survival curves with the TIGAR WT, TMU, Mut2 transduced mice in the CLP sepsis model.
- 3) Histological analyses of these mice should be included. The tissue sections showing expression of transduced TIGARs together with cell death and tissue damages would be highly informative.
- 4) The new Fig. S6b does not have any positive or negative controls.
- 5) The discussion "ubiquitination of NEMO prevents its binding with IKK leading to the activation of canonical IKK-NF- κ B signaling" is not accurate.
- 6) The discussion "ubiquitylation of RIPK1 provides multiple Ub scaffolds for the recruitment of TAB1/2 and activates TAK1" is not accurate.
- 7) Figure S11 legend needs to be fixed.
- 8) There is no justification why this study was done only in male mice.

Reviewer #2 (Remarks to the Author):

The authors have responded to my concerns about the mechanisms of how macrophages with re-expression of TIGAR in "dead mutants" regained their ability to cause intensified inflammation by macrophages.

REVIEWER COMMENTS

Reviewer #1 (Remarks to the Author):

The authors have added several new results in this revised version. The major addition is the in vivo lentivirus transduction to Tigar-deficient mice. This new result showed that wild type and the enzyme dead mutant but not the TAK1 binding lacking mutant restored CLP-induced inflammation in Tigar-deficient mice.

The additional results are intended to be sufficient to address the reviewers' concerns in physiological significance of the finding. However, the new results are not convincing due to missing critical controls and supporting data (see the major points below).

Overall, the revised manuscript does not sufficiently demonstrate the importance of the macrophage-derived metabolism-independent TIGAR regulation of inflammation in the in vivo setting.

Major points:

1) The new results using myeloid specific CD11b promoter-driven lentivirus of Flag-TIGAR, Flag-TMU or Flag-Mut2 should be very significant additions. However, the results are not convincing. It is known that myeloids are difficult cell types in lentivirus infection. The data showing the efficiency of transduction should have been included. What proportion of circulating and tissue myeloids express the transduced proteins? It should also have been confirmed whether other cell types did express the transduced TIGAR.

A: We would like to thank the reviewer for valuable comments. Per the indication, we determined the proportion of circulating and tissue myeloid cells expressing the

transduced proteins. When lentiviruses encoding Flag-TIGAR, TMU, or Mut2 were injected into *Tigar* KO mice, FACS analysis showed that average 37% of CD45⁺CD11b⁺ myeloid cells in blood and spleen expressed the transfected proteins (Fig. S11a-c). Western blot revealed that the transduced Flag-TIGARs were expressed in the isolated spleen F4/80⁺ macrophages instead of lung epithelial cells and hepatocytes, respectively (Fig. 7a, Fig. S11d). Immunohistochemical staining confirmed the expression of the transfected proteins in lung stroma (Fig. S11e).

2) The authors should present the mouse survival curves with the TIGAR WT, TMU, Mut2 transduced mice in the CLP sepsis model.

A: As suggested by the reviewer, we presented the mouse survival curves with the TIGAR, TMU, Mut2 transduced mice in the CLP sepsis model. Our results showed that the expression of TIGAR or TMU but not Mut2 reduced survival rate (Fig. 7c).

3) Histological analyses of these mice should be included. The tissue sections showing expression of transduced TIGARs together with cell death and tissue damages would be highly informative.

A: Per the suggestion of reviewer, we performed immunohistochemical staining of murine lung tissues. The positive staining of the transfected TIGARs in lung stroma are presented in (Fig. S11e), which is together with the results of impaired murine health conditions, decreased survival rates, exacerbated lung injury (Fig. S12a), and strong inflammation.

4) The new Fig. S6b does not have any positive or negative controls.

A: Thanks for the reviewer's kind suggestion. We used Takinib, an inhibitor of TAK1, as a positive control in the experiments. Our results showed that Takinib instead of TIGAR decreased TAK1 kinase catalytic activity (Fig. S6b).

5) The discussion "ubiquitination of NEMO prevents its binding with IKK leading to the activation of canonical IKK-NF- κ B signaling" is not accurate.

6) The discussion "ubiquitylation of RIPK1 provides multiple Ub scaffolds for the recruitment of TAB1/2 and activates TAK1" is not accurate.

A: We would like to thank the reviewer for the important indications. We re-edited the relevant part of discussion as follows: "It is known that the TAK1 kinase complex is recruited by ubiquitination of RIPK1 through the interaction between its polyubiquitin chains and TAB2. The polyubiquitin chains on RIPK1 also bind to NEMO, resulting in recruitment of the IKK complex. IKK β in the complex is then phosphorylated and activated by TAK1, leading to the phosphorylation, ubiquitination, and degradation of I κ B. NF- κ B subsequently translocates into the nucleus to activate gene expression^{1,2}." in lines 356-361.

7) Figure S11 legend needs to be fixed.

A: We would like to thank the reviewer for pointing out this mistake and have corrected it in the revised manuscript.

8) *There is no justification why this study was done only in male mice.*

A: The reason using male mice in this study is based on the discovery that estrogen influences the survival and inflammation in septic mice³. Per the indication, we added a sentence in Animal models, Methods section. “Male mice were used in this study to avoid possible influence of estrogen on survival and inflammation in septic mice³.” (line 405-406).

Reviewer #2 (Remarks to the Author):

The authors have responded to my concerns about the mechanisms of how macrophages with re-expression of TIGAR in “dead mutants” regained their ability to cause intensified inflammation by macrophages.

A: We would like to thank the reviewer for positive comments.

References:

1. Ea, C.K., Deng, L., Xia, Z.P., Pineda, G. & Chen, Z.J. Activation of IKK by TNFalpha requires site-specific ubiquitination of RIP1 and polyubiquitin binding by NEMO. *Mol Cell* **22**, 245-257 (2006).
2. Clucas, J. & Meier, P. Roles of RIPK1 as a stress sentinel coordinating cell survival and immunogenic cell death. *Nat Rev Mol Cell Biol* **24**, 835-852 (2023).
3. Diodato, M.D., Knoferl, M.W., Schwacha, M.G., Bland, K.I. & Chaudry, I.H. Gender differences in the inflammatory response and survival following haemorrhage and subsequent sepsis. *Cytokine* **14**, 162-169 (2001).

REVIEWERS' COMMENTS

Reviewer #1 (Remarks to the Author):

The authors have addressed all my concerns. The results collectedly demonstrate TIGAR's novel role in innate immune responses.

REVIEWER COMMENTS

Reviewer #1 (Remarks to the Author):

The authors have addressed all my concerns. The results collectedly demonstrate TIGAR's novel role in innate immune responses.

A: We would like to thank the reviewer for positive comments.